# The interplay between biomolecular assembly and phase separation

**Giacomo Bartolucci[1,2]\*, Ivar S Haugerud[2], Thomas CT Michaels[3], Christoph A Weber[2]\***

[1]Max Planck Institute for the Physics of Complex Systems, Dresden, Germany; [2]Faculty of Mathematics, Natural Sciences, and Materials Engineering: Institute of Physics, University of Augsburg, Augsburg, Germany; [3]ETH Zurich, Institute of Biochemistry, Zurich, Switzerland

## eLife Assessment

The authors present an **important** theoretical framework that describes the interplay between liquid-liquid phase separation and protein aggregation within a mean-field model. This work will be of high interest to the biophysics and molecular biology communities, as it will help understand and analyse assembly within biomolecular condensates in cells or in-vitro. Major strengths of this **convincing** work are the consideration of aggregates with various dimensionality and the possibility for protein gelation.

**\*For correspondence:**
bartolucci@ub.edu (GB);
christoph.weber@physik.uni-augsburg.de (CAW)

**Competing interest:** The authors declare that no competing interests exist.

**Abstract** Many biological functions and dysfunctions rely on two fundamental processes, molecular assembly and the formation of condensed phases such as biomolecular condensates. Condensed phases generally form via phase separation, while molecular assemblies are clusters of molecules of various sizes, shapes, and functionality. We developed a theory that relies on thermodynamic principles to understand the interplay between molecular assembly and phase separation. We propose two prototypical classes of protein interactions and characterize their different equilibrium states and relaxation dynamics. We obtain results consistent with recent in vitro experimental observations of reconstituted proteins, including anomalous size distribution of assemblies, the gelation of condensed phases, and the change in condensate volume during ageing. Our theory provides the framework to unravel the mechanisms underlying physiological assemblies essential for cellular function and aberrant assemblies which are associated with several neurodegenerative disorders.

## Introduction

Due to their structural complexity, proteins can interact in different ways, leading to coexisting phases or assemblies such as fibers and aggregates. Long-lived assemblies are often kept together by strong adhesive forces, with binding free energies ranging from 9 $k_{\mathrm{B}}T$ in the case of insulin dimers (**Doty and Myers, 1953**), over 2.5 $k_{\mathrm{B}}T$ per beta-sheet in amyloid fibers, to the 0.9 $k_{\mathrm{B}}T$ per beta-sheet in the formation of assemblies of specific FUS segments called low-complexity aromatic-rich kinked segments (**Hughes et al., 2018**). Weak interactions are often responsible for the separation into liquid phases, each of distinct molecular compositions. The interaction free energies associated with the formation of P granules via phase separation in living cells are about 0.5 $k_{\mathrm{B}}T$ per molecule (**Fritsch et al., 2021**). The biological function of both assemblies and phase-separated compartments relies on the recruitment of specific biomolecules such as proteins, RNA, or DNA (**Banani et al., 2017**; **Nüske et al., 2020**; **Zbinden et al., 2020**; **Wang et al., 2021**). Since assemblies and condensed phases

can adhere to membrane surfaces, both not only mediate mechanisms for sorting and transport of molecules (*Alberti et al., 2019*) but also affect the composition, shape and properties of intra-cellular surfaces (*Zhao and Zhang, 2020*; *Agudo-Canalejo et al., 2021*; *Babl et al., 2022*; *Lu et al., 2022*).

Despite these similarities, molecular assemblies and coexisting phases also exhibit crucial differences. While the size of a condensed phase at equilibrium increases with the size of the system (*Weber et al., 2019b*), this is not necessarily the case for molecular assemblies (*Stockmayer, 1943*; *Krapivsky et al., 2010*; *Israelachvili, 2015*; *Safran, 2019*). Moreover, the assembly kinetics tends to an equilibrium between assemblies of different sizes (*Stockmayer, 1943*; *Krapivsky et al., 2010*; *Israelachvili, 2015*; *Hagan and Grason, 2021*), while condensed phases equilibrate the physico-chemical properties such as temperature, pressure, and chemical potential between the spatially separated phases (*Weber et al., 2019b*). These differences suggest a rich interplay in a system where the molecular constituents can both oligomerise forming assemblies and give rise to coexisting phases (*Dumetz et al., 2008*; *Shin and Brangwynne, 2017*; *McCall et al., 2018*; *Wiegand and Hyman, 2020*; *Liu et al., 2020*; *Küffner et al., 2021*).

In the last years, the interplay between phase separation and assembly formation has been the focus of many experimental efforts. Different proteins capable of forming condensed phases were shown to form oligomers below the saturation concentration (*Beutel et al., 2019*; *Kar et al., 2022*). The authors proposed that such oligomers affect the phase separation propensity; however, the detailed mechanism remains elusive. Moreover, several experimental studies indicate that proteins in the protein-rich phase are linked, reminiscent of a physical gel (*Li et al., 2012*; *Franzmann et al., 2018*; *Guillén-Boixet et al., 2020*). Molecular simulations were performed that aimed at the sequence-specific origin of such phenomena (*Harmon et al., 2017*; *Martin et al., 2020*; *Rana et al., 2021*; *Pappu et al., 2023*). However, even in elegantly coarse-grained simulation approaches, the large number of parameters makes it difficult to extract general mechanisms across different proteins. To develop an understanding of such general mechanisms that underlie the interplay between phase separation and molecular assembly, a theoretical framework that relies on thermodynamic principles is lacking.

While the theory of phase separation of a low number of different components (*Bray and Rutenberg, 1994*; *Weber et al., 2019b*), as well as the formation of molecular assemblies in dilute environments (*Smoluchowski, 1917*; *Stockmayer, 1943*; *Oosawa and Asakura, 1975*), are well developed, only a few works addressed assembly formation beyond the dilute limit, where assemblies can form and also phase separate. For example, it has been shown that, in the presence of coexisting phases, the assembly size distributions at equilibrium can vary in the two phases (*Blankschtein et al., 1985*; *Li et al., 2023*) and that the protein-rich phase can gelate (*Semenov and Rubinstein, 1998*; *Zilman et al., 2003*; *Tanaka, 2011*; *Deviri and Safran, 2020*). These studies account for the scaling of the internal free energies of assemblies with their size but neglect the size dependence of the interaction propensities. Moreover, a discussion of the coupled phase separation and assembly kinetics is lacking. Other authors focused on systems composed of a scaffold component that drives phase separation, and studied the dilute assembly kinetics of a second component that can interact with the scaffold (*Weber et al., 2019a*; *Michaels et al., 2019*; *Pönisch et al., 2023*; *Hagan and Mohajerani, 2023*). In these works, the assemblies are considered to be dilute, and the feedback of the assembly kinetics on the phase-separated compartment is neglected.

In this work, we introduce a framework that unifies the thermodynamic theories for phase separation with the theories developed for the formation of micelles and molecular assemblies at dilute conditions. This multi-scale framework bridges assembly, a phenomenon occurring at the molecular scale, with phase separation occurring in the macroscopic realm. We present two classes of size-dependent interactions that are inspired by biologically relevant proteins. Our theory is able to reproduce results observed in recent experimental studies, such as the emergence of anomalous size distribution below saturation and the gelation of condensed phases above saturation, and characterise for which class and parameter values these phenomena manifest. Furthermore, we propose a non-equilibrium thermodynamic theory for the kinetics of molecular assembly at non-dilute conditions which can lead to macroscopic, condensed phases above the saturation concentration. The complexity of our theory is reflected in a high-dimensional phase space that is set by the number of differently sized assemblies. We developed efficient numerical schemes to investigate the kinetics of such systems for the case where diffusion is fast compared to assembly kinetics. In particular, we study how condensates, initially

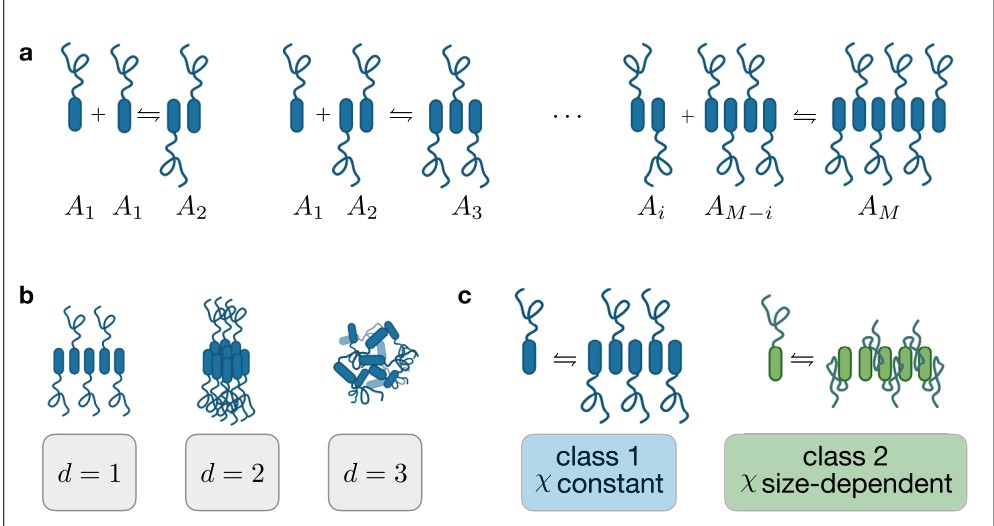

**Figure 1.** Illustration of assembly reaction scheme and classification. (**a**) Illustration of the chemical reaction network associated with the formation of assemblies $A_i$ with size $i$. (**b**) Identification of three classes based on assembly dimension: d=1, 2, 3. (**c**) Classification of assemblies based on the scaling of their Flory-Huggins interaction propensity.

formed via the phase separation of monomers from the solvent change in response to the formation of assemblies. Our unified theory provides the answer to crucial biological questions, such as under which conditions the presence of coexisting phases affects the formation of assemblies and could be key to interpreting and understanding recent observations of protein condensation in vitro (*Kanaan et al., 2020*), and in the cell cytoplasm (*Guillén-Boixet et al., 2020*; *Ray et al., 2020*; *Kar et al., 2022*; *Seim et al., 2022*).

## Results

### Assembly and phase equilibria

We begin by reviewing the equilibrium theory of multi-component mixtures composed of solvent (s) and monomers ($i = 1$) that can form assemblies composed of $i$ monomers, with $i < M$ see *Figure 1a*. We consider a maximum assembly size $M$, but, as we will see, this assumption must be relaxed when monomers tend to form an assembly of infinite size. In the case when monomers and assemblies are dissolved in the solvent, the free energy density of the solution can be written as *Blankschtein et al., 1985*; *Flory, 1942*; *Huggins, 1942*; *Tanaka, 2011*:

$$
\begin{aligned}
f_{\text{sol}} \quad &= \frac{k_{\text{B}}T}{\nu_{\text{s}}} \left( \sum_{i=1}^{M} \frac{\phi_i}{\rho_i} \ln\left(\frac{\phi_i}{\rho_i}\right) + \frac{\omega_i}{k_{\text{B}}T}\phi_i + \sum_{i\neq j=1}^{M} \frac{\chi_{ij}}{2k_{\text{B}}T}\phi_i\phi_j \right. \\
&\quad \left. + \phi_{\text{s}} \ln \phi_{\text{s}} + \frac{\omega_{\text{s}}}{k_{\text{B}}T}\phi_{\text{s}} + \sum_{i=1}^{M} \frac{\chi_{i\text{s}}}{k_{\text{B}}T}\phi_i\phi_{\text{s}} \right),
\end{aligned}
\tag{1}
$$

where $\rho_i = \nu_i/\nu_{\text{s}}$ denotes the relative molecular volume with $\nu_i$ as the molecular volume of assembly of size $i$, and $\nu_{\text{s}}$ denotes the solvent molecular volume. The solvent volume fraction can be expressed as a function of the assembly volume fractions via $\phi_{\text{s}} = 1 - \sum_{i=1}^{M} \phi_i$. The first and fourth terms in *Equation 1* are the mixing entropies. The second and fifth contributions of $f_{\text{sol}}$ account for the internal free energies $\omega$. Here, $\omega_{\text{s}}$ denotes the internal free energy of the solvent, and $\omega_i$ are the internal free energies per monomer of an assembly of size $i$, stemming from the free energy of internal bonds that lead to assembly formation. Note that we chose to keep $\phi_i/\rho_i$ in the logarithm argument instead of reabsorbing the linear term $-\phi_i \ln(\rho_i)/\rho_i$ in the internal free energies $\omega_i$. With this choice, $\omega_i$, depends only on the free energies of the bonds, see Appendix 2, (*Flory, 1942*), and the recent overview in the SI of *Pönisch et al., 2023*. The third and last terms in *Equation 1* capture the interactions of monomers belonging to different assemblies and with the solvent, where $\chi_{ij}$ is the corresponding interaction parameter. We note that varying the temperature $T$ affects the contributions of both the interaction

and the internal free energy terms. The exchange chemical potentials of monomers belonging to an assembly of size $i$ reads

$$\mu_i = \nu_1 \frac{\partial f_{\text{sol}}}{\partial \phi_i}.$$ (2)

## Assembly equilibrium

Assemblies can grow and shrink via association and dissociation. Such transitions among assemblies of different sizes are reminiscent of chemical transitions, see *Figure 1a*. The condition of chemical equilibrium reads (*Israelachvili, 2015*):

$$\mu_i = \mu_1 = \text{const.} \quad \forall i = 2 \ldots M,$$ (3)

where $\mu_i$ is the exchange chemical potential of monomers belonging to an assembly of size $i$; see *Equation 2*. Using the free energy *Equation 1* and the equilibrium conditions *Equation 3*, we can express the volume fraction of the assembly of size $i$ as a function of the monomer volume fraction $\phi_1$ in the form:

$$\phi_i = \rho_i \left(\frac{\phi_1}{\rho_1}\right)^{\rho_i/\rho_1} \exp\left[\rho_i \left(1 - \frac{\omega_i - \omega_1}{k_B T} - \frac{\chi_{is} - \chi_{1s}}{k_B T}\phi_s - \sum_j \frac{\chi_{ij} - \chi_{1j}}{k_B T}\phi_j\right) - 1\right].$$ (4)

The equation above, together with the conservation of monomers

$$\phi_{\text{tot}} = \sum_{i=1}^{M} \phi_i,$$ (5)

allows us to rewrite the volume fraction $\phi_i$ of each assembly of size $i$, as a function of the conserved quantity $\phi_{\text{tot}}$. This relation $\phi_i = \phi_i(\phi_{\text{tot}})$ has an analytical expression in the case $d = 1$, see *Equation 24 and Equation 26* in Appendix 3.

## Phase equilibrium

Two phases in an incompressible, multi-component system are at phase equilibrium when the chemical potentials $\mu_i$ and the osmotic pressure $\Pi = -f_{\text{sol}} + \sum_{i=1}^{M} \phi_i \partial f_{\text{sol}}/\partial \phi_i$ balance in each phase (*Safran, 2019*; *Weber et al., 2019b*):

$$\mu_i^{\text{I}} = \mu_i^{\text{II}},$$ (6a)

$$\Pi^{\text{I}} = \Pi^{\text{II}},$$ (6b)

where the superscripts I and II indicate the $\phi_{\text{tot}}$-rich and II the $\phi_{\text{tot}}$-poor phase, respectively.

## Thermodynamic equilibrium

Our system is at thermodynamic equilibrium when assembly and phase equilibrium hold simultaneously. The conditions above for phase equilibrium can thus be rewritten using $\phi_i(\phi_{\text{tot}})$ *Equation 4*. In particular, the free energy density *Equation 1* can be recast in terms of the conserved variable, $\phi_{\text{tot}}$ (*Adame-Arana et al., 2020*; *Bartolucci et al., 2021*). The phase diagram of the system can then be obtained via the common tangent construction (i.e. Maxwell construction). This construction corresponds to the balance between the exchange chemical potentials and the osmotic pressure in both phases; see Chapter 2 in *Adame-Arana et al., 2020*. *Bartolucci et al., 2021*:

$$\mu(\phi_{\text{tot}}^{\text{I}}) = \mu(\phi_{\text{tot}}^{\text{II}}),$$ (7a)

$$\mu(\phi_{\text{tot}}^{\text{I}}) = \frac{f_{\text{sol}}(\phi_{\text{tot}}^{\text{II}}) - f_{\text{sol}}(\phi_{\text{tot}}^{\text{I}})}{\phi_{\text{tot}}^{\text{II}} - \phi_{\text{tot}}^{\text{I}}}.$$ (7b)

*Equation 3*, *Equation 7a*, *Equation 7b* establish how the behaviour of the mixture at equilibrium is affected by the parameters of the free energy in *Equation 1*, such as internal free energies $\omega_i$ or

interaction parameters $\chi_{ij}$. In the next section, we introduce classes based on the scaling of such parameters with assembly size $i$.

## Scaling of molecular volumes, internal free energies and interaction energies with assembly size

The composition of the phase-separated compartments and the size distributions of the assemblies in each phase will depend on the scaling form of the key parameters of the model with the assembly size $i$: the relative molecular volumes ($\rho_i$), the internal free energy of assemblies ($\omega_i$), and the interaction energies of assemblies among themselves ($\chi_{ij}$), and with the solvent ($\chi_{is}$).

In this work, we choose $\rho_i = i$. This choice reflects the fact that no solvent is present in assemblies and that the chemical reaction network in *Figure 1a* conserves the sum of molecular volumes. The assumption $\rho_i = 1$, leads to a phase diagram that is symmetric about $\phi_1 = 1/2$, if only monomers and solvent are present. This might seem an oversimplification, but once assemblies of different sizes form, the phase diagram becomes asymmetric, as expected in most biological applications. Thus, this assumption simplifies the framework, while the equilibrium states retain the essential qualitative features of realistic systems. In our model, assemblies form as a consequence of internal bonds among monomers. Each bond is associated with a free energy $e_{int} - s_{int}T$, with $e_{int}$ and $s_{int}$ the enthalpic and an entropic contribution, respectively. In Appendix 2, we derive the scaling relationships for the internal free energies of linear ($d = 1$), planar ($d = 2$) and three-dimensional ($d = 3$) assemblies:

$$\omega_i \simeq \omega_\infty - \frac{e_{int} - s_{int}T}{i^{1/d}}. \tag{8}$$

The physical origin of the dependency $i^{1/d}$ is the scaling of the number of internal bonds in an assembly of dimension $d$. In *Equation 8*, $\omega_\infty = \lim_{i \to \infty} \omega_i$ is a constant that does not affect chemical nor phase equilibrium, except in the limit $M \to \infty$, which will be discussed later. In Appendix 5, we discuss how variations of bond energy affect phase separation.

For the scaling of interaction energies $\chi_{ij}$ and $\chi_{is}$, we introduce two classes inspired by biologically relevant classes of proteins that can form assemblies and phase separate:

1. Class 1: Constant assembly-solvent interactions. This class corresponds to the case where each monomer, independently of the assembly it is part of, interacts equally with the solvent $\chi_{is} = \chi$. Moreover, monomers in assemblies of different sizes interact equally with each other, implying that the corresponding Flory-Huggins parameter $\chi_{ij}$ vanishes:

$$\chi_{is} = \chi, \qquad \chi_{ij} = 0, \tag{9}$$

for a derivation of this relation starting from a lattice model, see Appendix 2. This class is inspired by biologically relevant proteins for which the oligomerisation domains are well separated along the protein from hydrophobic phase separation domains. In this case, when monomers form an assembly, their phase separation domains remain exposed, leading to a monomer-solvent interaction that does not depend on assembly size. Examples belonging to this class include synthetic constructs like the so-called 'Corelets' (*Bracha et al., 2018*), realised tethering intrinsically disordered protein fragments to oligomerizing domains (*Bracha et al., 2018*), and proteins like NPM1, whose N-terminal oligomerisation domain (that allows for the formation of pentamers) is considered to be separated from the disordered region (responsible for phase separation) and the RNA binding domain (*Mitrea et al., 2014*; *Feric et al., 2016*).

2. Class 2: Size-dependent assembly-solvent interactions. This class describes the case where monomers in the assembly bulk and monomers at the assembly boundary have different interaction propensies with the solvent ($\chi'$ and $\chi$, respectively, see Appendix 2 for details). Similar to class 1, monomers in assemblies of different sizes interact equally with each other, leading to

$$\chi_{is} = \chi' + \frac{\chi - \chi'}{i^{1/d}}, \quad \chi_{ij} = 0. \tag{10}$$

The dependency $i^{1/d}$ originates from the scaling of the number of monomers in the bulk and in the boundary of assemblies, in different spatial dimensions $d$. This class corresponds to the general case in which the oligomerisation domains of protein overlap with the phase

separation domains. This case applies to segments of the intrinsically disordered region of the protein FUS, for example. In fact, recent experiments have shown the formation of assemblies in solutions containing specific FUS domains, called low-complexity aromatic-rich kinked segments (LARKS) (*Luo et al., 2018*; *Hughes et al., 2018*). Strikingly, it was shown that hydrophobic domains along LARKS were buried in the formation of these assemblies, and the author could quantify the hydrophobic area buried upon assembly formation. Another example could be Whi3, since it has been recently found that mutation that enhances oligomerisation strength lowers the density of Whi3 in the RNP condensates (*Seim et al., 2022*), suggesting that the formation of assemblies could screen Whi3 phase separation propensity. Finally, the formation of DNA nanostars has been recently shown to inhibit phase separation in DNA liquids (*Hegde et al., 2024*).

We consider the relevant interaction parameters, like internal free energies $\omega_i$ and interaction propensities $\chi$ and $\chi'$ as control parameters and vary them independently. This is a simplification since, in biology, they might be coupled, e.g., the swelling of the intrinsically disordered regions could lead to variations in monomer binding strength (*Connor et al., 2022*). In the next sections, we characterize the equilibrium behaviour of systems belonging to these classes.

## Assembly size distributions below and above saturation

Here, we discuss the differences between assembly equilibrium in homogeneous and phase-separating systems and outline the implications for biological mixtures. We first consider systems that are spatially homogeneous and composed of linear assemblies ($d = 1$). Homogeneity can be realized in dilute solutions if the total protein volume fraction $\phi_{\text{tot}}$ is below the saturation volume fraction of phase separation $\phi_{\text{tot}}^{\text{II}}(T)$ (for a definition, see Assembly and phase equilibria). Homogeneous systems governed by *Equation 4* at equilibrium, obeying the conservation *Equation 5*, exhibit two limiting behaviours depending on the value of the conserved variable $\phi_{\text{tot}}$. We define the *assembly threshold* $\phi^*(T)$, that separates these two behaviours, as the value of $\phi_{\text{tot}}$ for which there is a maximum of $\phi_i$ for monomers ($i = 1$) with zero slope:

$$\left.\frac{\partial \phi_i(\phi_{\text{tot}})}{\partial i}\right|_{i=1,\phi^*} = 0. \tag{11}$$

Indeed, for $\phi_{\text{tot}} \ll \phi^*$ the size distribution of linear assemblies ($d = 1$) is dominated by monomers ($\phi_1 \simeq \phi_{\text{tot}}$) while larger assemblies have vanishing volume fraction. For higher total volume fractions ($\phi_{\text{tot}} \gtrsim \phi^*$), the monomer volume fraction saturates at $\phi_1 \lesssim \phi^*$ and bigger assemblies start to populate the mixture. Above $\phi^*$, the distribution becomes peaked at a value $i_{\text{max}} > 1$ and then exponentially decays for larger $i$; see *Appendix 3—figure 1* in Appendix 2. Both the maximum and the average of the distribution $\phi_i$ scale with $\sqrt{\phi_{\text{tot}}}$ indicating that as $\phi_{\text{tot}}$ is increased, larger assembly populate the system; see Appendix 3 for a detailed discussion for Class 1.

Now we consider systems that can phase separate. As outlined in Assembly and phase equilibria, at assembly equilibrium, we can recast the free energy as a function of the conserved variable $\phi_{\text{tot}}$ by using *Equation 4*. For sufficiently large assembly-solvent interaction parameters $\chi$ and $\chi'$, the system can demix into two phases with different total volume fractions $\phi_{\text{tot}}^{\text{I}}$ and $\phi_{\text{tot}}^{\text{II}}$, which are the solutions of *Equation 7a*, *Equation 7b* . By means of $\phi_{\text{tot}}^{\text{I/II}}$, we can calculate the whole assembly size distribution in the two phases, i.e., $\phi_i^{\text{I/II}}$, via *Equation 4; Equation 5*.

We now present numerical results obtained via the codes available at https://github.com/Giaco-Barto/AssemblyAndPhaseSeparation (copy archived at *Bartolucci, 2026*). We first discuss linear assemblies belonging to class 1, in the regime of high assembly strength $-e_{\text{int}}/\chi \gg 1$; see *Figure 2a–c*. In *Figure 2a*, we show the corresponding phase diagram as a function of $\phi_{\text{tot}}$ and the rescaled temperature $T/T_0$ with $T_0 = \chi/k_{\text{B}}$. The domain enclosed by the binodal corresponds to phase separation. As indicated by the colour code (depicting the monomer fraction $\phi_1/\phi_{\text{tot}}$) each point in the diagram can have different assembly composition. In green, we plot the assembly threshold $\phi^*(T)$, at which intermediate-sized assemblies start to appear. Note that, with this choice of parameters, the assembly threshold precedes in $\phi_{\text{tot}}$ the dilute branch of the binodal. We stress that, for $d = 1$, crossing the assembly threshold does not lead to a phase transition since, in contrast to crossing the binodal, it is not accompanied by a jump in the free energy or its derivatives. We can now define regions corresponding to qualitatively different phase and assembly behaviour. In particular, starting from a homogeneous system composed of monomers only (region 'i'), increasing $\phi_{\text{tot}}$ leads to the emergence of

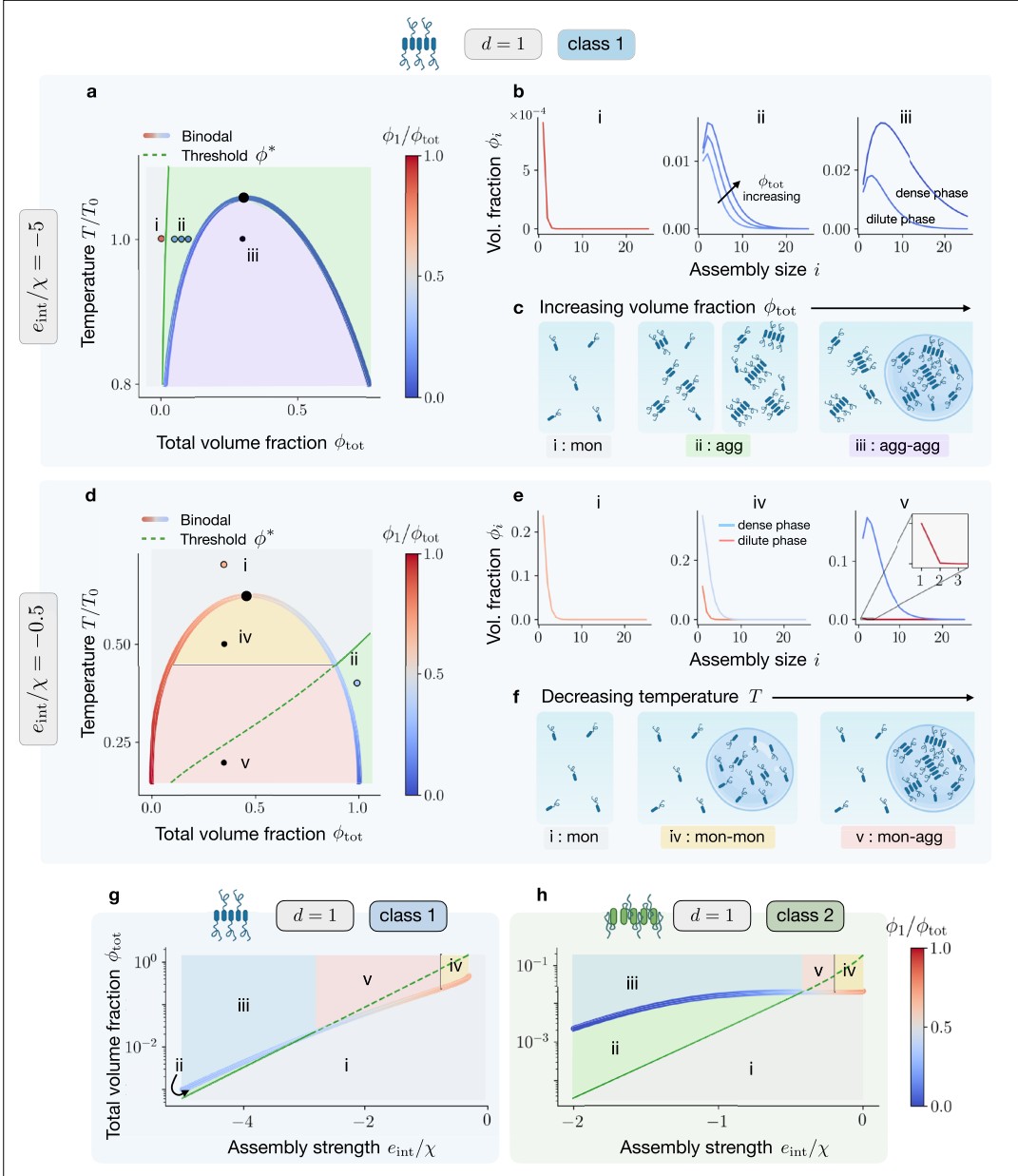

**Figure 2.** Phase diagram and assembly size distributions for different classes and assembly strengths. (**a**) Phase diagram as a function of $\phi_{tot}$ and rescaled temperature $T/T_0$ (with $T_0 = \chi/k_B$) in the regime of high assembly strength, i.e., $-e_{int}/\chi \gg 1$. The green line is the volume fraction threshold $\phi^*(T)$ at which intermediate-sized assemblies start to appear, which in this regime precedes the binodal (coloured curve). As indicated by the colour code, the monomer fraction $\phi_1/\phi_{tot}$ mildly varies in the two phases. (**b**) Size distributions and c pictorial representations corresponding to different regions of the phase diagram, defined by the relative position of the binodal and the assembly threshold. In region 'i,' the system is homogeneous and composed of monomers only. Increasing the total volume fraction of assemblies $\phi_{tot}$ beyond the assembly threshold $\phi^*$, the system enters region 'ii' where intermediate assemblies appear. Here, the sizes corresponding to the maximum and the average of the distribution $\phi_i$ scale with $\sqrt{\phi_{tot}}$, see Appendix 3. Finally, once $\phi_{tot}$ exceeds the binodal, the system enters region 'v' and demixes in two phases, both rich in intermediate assemblies. In (**d-f**), we focus on the low assembly strength regime, i.e., $-e_{int}/\chi \sim 1$. In phase diagram d, the binodal now precedes in $\phi_{tot}$ the assembly threshold. e In region 'iv,' the system phase separates but in both phases monomers dominate the size distribution, while in region 'v' the $\phi_{tot}$-rich phase becomes populated by intermediate-sized assemblies. Progressively lowering the temperature allows switching between these regions, as depicted in (**f**). (**g, h**) Behaviour of dilute mixtures as a function of assembly strength, for the two different classes. Notably, assembly below saturation becomes much more accessible for class 2, as can be seen by comparing the green regions 'ii' in (**g** and **h**).

intermediate-sized assemblies (region 'ii'). Increasing $\phi_{tot}$ further, the system demixes into two phases, both of which are rich in intermediate assemblies (region 'iii'). Representative size distributions and illustrations of the state of the systems in the different regions are shown in *Figure 2b and c*, respectively. For parameter values, see *Appendix 1—table 1* in Appendix 1. This analysis showcases the potential of this framework to describe the appearance of mesoscopic clusters below the saturation, as recently observed experimentally in *Kar et al., 2022*.

Remaining within class 1, we now discuss the case of low assembly strength $-e_{int}/\chi \sim 1$; see *Figure 2d–f*. The interception between the binodal and the assembly threshold $\phi^*$ defines two new regions, 'iv' and 'v,' see *Figure 2d*. In particular, in region 'iv,' both binodal branches lie below the assembly threshold, resulting in monomers dominating both coexisting phases, see *Figure 2e*, centre. On the other hand, in region 'v' the $\phi_{tot}$-rich phase exceeds the assembly threshold, resulting in phases with dramatically different compositions: the $\phi_{tot}$-poor phase is populated only by monomers while intermediate-sized assemblies develop in the $\phi_{tot}$-rich phase, see *Figure 2e* right. The spatial separation of assemblies into the $\phi_{tot}$-rich phase has likely far-reaching biological implications. For example, it may reduce the toxic effects of aggregates in neurodegenerative diseases (*Zbinden et al., 2020*). In *Figure 2f*, we illustrate states corresponding to fixed $\phi_{tot}$ and decreasing temperature $T$. Starting from a homogeneous monomeric state, region 'i,' the system transitions into a demixed state with monomers dominating both phases, region 'iv,' and finally to a demixed state with larger assemblies abundant in the $\phi_{tot}$-rich phase, region 'v.'

We now highlight the differences between the two classes defined in Scaling of molecular volumes, internal free energies and interaction energies with assembly size. In particular, we characterise how mixtures of monomers prone to assembly and phase separation behave with increasing $\phi_{tot}$, varying the assembly strength $e_{int}/\chi$ but keeping the temperature $T$ fixed. In particular, for class 1, the emergence of assemblies before saturation typically occurs for a very narrow interval of volume fractions, see the green region labelled with 'ii' in *Figure 2g*. Strikingly, for class 2, assembly below saturation are more favoured; see again region 'ii' in *Figure 2h*. This difference arises because, within class 2, monomers in the bulk of an assembly have reduced interaction propensity with respect to the boundary ones. As a consequence, the formation of large clusters shifts the onset of phase separation to higher $\phi_{tot}$ values. Summing up, phase separation controls the onset (see also Appendix 5) and localisation of assemblies. For the considered parameters, the $\phi_{tot}$-rich phase contains larger assemblies compared to the $\phi_{tot}$-poor phase. In the next section, we will see that, for planar and spherical assemblies ($d > 1$), this difference can become even more extreme with the protein-rich phase becoming one giant assembly, also referred to as the gel phase (*Krapivsky et al., 2010*).

## Gelation of the protein-rich phase

In this section, we discuss the case of planar ($d = 2$) and three-dimensional assemblies, ($d = 3$), referring for simplicity to systems belonging to Class 1. In this case, as shown in Appendix 4, even when neglecting protein solvent interactions ($\chi = 0$), the system can undergo a transition where the protein-rich phase becomes a large (macroscopic) assembly. In the thermodynamic limit $M \to \infty$, this transition corresponds to a phase transition, i.e., allowing for the emergence of an infinitely large assembly. In fact, above the threshold volume fraction $\phi^{sg}$, we observe the emergence of such a macroscopic assembly occupying a finite fraction of the system volume that contains a macroscopic fraction of all monomers in the system – a behaviour reminiscent of Bose-Einstein condensation; see for example, Chapter 7.3 of *Tanaka, 2011* for an interesting discussion on this analogy. The threshold volume fraction $\phi^{sg}$ is affected by temperature and the free energy of internal bonds $\Delta\omega$ *Equation 31*; the definition of $\Delta\omega$ is given in *Equation 21*. We call this macroscopic assembly the gel phase, in agreement with previous literature (*Semenov and Rubinstein, 1998*; *Zilman et al., 2003*; *Tanaka, 2011*; *Deviri and Safran, 2020*). Please note that, since we do not explicitly includes the solvent in assembly formation (see reaction scheme in *Figure 1a*), in our model the gel corresponds to a phase without solvent, $\phi_{tot} = 1$. To account for biological gels that can be rich in water, our theory can be straightforwardly extended by incorporating the solvent into the reaction scheme.

We now focus on systems that phase separate as the result of interactions with the solvent ($\chi \neq 0$ in *Equation 9*) and discuss the interplay between phase separation and gelation. Volume fractions in the coexisting phases are determined by Equation 7 and assembly equilibrium requires that *Equation 3* be satisfied. As pointed out in Sec. Assembly and phase equilibria, we aim to find an expression for

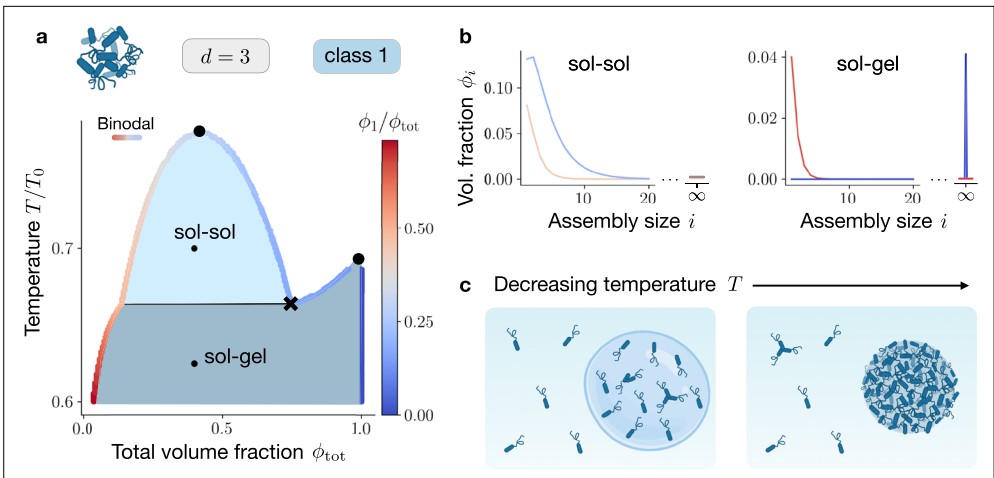

**Figure 3.** Gelation transition in phase-separating systems. (**a**) Phase diagram for planar (d=2) and three-dimensional (d=3) assemblies in the limit $M \to \infty$, as a function of $\phi_{\text{tot}}$ and the rescaled temperature $T/T_0$ (with $T_0 = \chi/k_{\text{B}}$). The coloured curve represents the binodal associated with the free energy $f$, which accounts for the emergence of an infinite assembly. The colour code of the binodal line depicts the monomer fraction $\phi_1/\phi_{\text{tot}}$ in the phases, and the black cross 'x' indicates the triple point. In the region labelled as 'sol-sol,' the system demixes into two phases both populated mainly by monomers, see panel (**b**), with $\phi_{\text{tot}}^{\text{I/II}} < 1$. In the region labelled as 'sol-gel,' on the other hand, a phase (the 'sol'), obeying $\phi_{\text{tot}}^{\text{sol}} < 1$, coexists with a phase (the 'gel') that is a macroscopic assembly, containing no solvent ($\phi_{\text{tot}}^{\text{sol}} = 1$). The latter scenario is represented in panel (**b**), right side. (**c**) Lowering the temperature allows transitions from the 'sol-sol' to the 'sol-gel' region, which manifest with a jump in the total volume fraction of the protein-rich phase.

---

$\phi_i(\phi_{\text{tot}})$ via **Equation 3; Equation 5**, and then substitute it into the free energy **Equation 1**. However, for planar ($d = 2$) and three-dimensional assemblies, ($d = 3$), performing the thermodynamic limit $M \to \infty$ leads to a free energy composed of series that diverges in the thermodynamic limit. We know that this divergence is physical, and is caused by the gelation transition. The divergence can be resolved by introducing explicitly a term in the free energy that accounts for an infinite-sized assembly – the gel. Thus, we write the system free energy as a composition of the solution free energy $f_{\text{sol}}$ and the gel free energy $f_{\text{gel}}$:

$$f = f_{\text{sol}} + f_{\text{gel}}, \tag{12}$$

where $f_{\text{sol}}$ is defined in **Equation 1**. The gel free energy reads

$$f_{\text{gel}} = \frac{\omega_\infty}{\nu_1} \, \delta \left( 1 - \phi_{\text{tot}} \right), \tag{13}$$

with $\delta(\cdot)$ denoting the delta distribution. The gel free energy $f_{\text{gel}}$ is the free energy of a state with no solvent, where all monomers belong to an assembly of size $i \to \infty$. In fact, in the limit $\phi_i = 0$ for all finite $i$ and $\phi_{\text{tot}} = 1$, the free energy in **Equation 1** simplifies to the single contribution $\omega_\infty/\nu_1$. This observation sheds light on $\omega_\infty$, which has the physical interpretation of free energy associated with each bond among monomers belonging to the gel. For this reason, we chose $\omega_\infty$ to be proportional to the bond free energy among monomers in solution ($e_{\text{int}} - Ts_{\text{int}}$); see Appendix 4 for more details.

We can now perform a Maxwell construction by using **Equation 12** in **Equation 7a**, **Equation 7b**. The resulting phase diagram is displayed in **Figure 3a**, where the binodal is coloured by the monomer fraction $\phi_1/\phi_{\text{tot}}$ in the coexisting phases. In phase-separated systems, gelation can be considered as a special case of phase coexistence between a protein-poor phase ('sol'), in which $\phi^{\text{sol}} < 1$, and the gel phase, corresponding to $\phi^{\text{gel}} = 1$. The domain in the phase diagram where a gel phase coexists with a soluble phase is shaded in blue and labelled as 'sol-gel' in **Figure 3a**. In the same panel, we show that lowering the temperature for large $\phi_{\text{tot}}$ leads to a transition from the homogeneous state to the sol-gel coexistence. By contrast, for intermediate volume fractions, the system transits first through a domain corresponding to two-phase coexistence; see light blue domain labelled as 'sol-sol' in **Figure 3a**, where $\phi_{\text{tot}} < 1$ in both phases. At the triple point (marked with the black cross), the gel phase of volume

fraction $\phi_{\text{tot}} = 1$ coexists with two 'sol' phases, for which $\phi_{\text{tot}} < 1$. In **Figure 3b**, we show assembly size distributions representative of the 'sol-sol' and 'sol-gel' regions. The transition from the 'sol-sol' to the 'sol-gel' region is accompanied by a jump in the total volume fraction of the protein-rich phase $\phi_{\text{tot}}^{\text{I}}$, while the value in the protein-poor phase $\phi_{\text{tot}}^{\text{II}}$ changes smoothly. This finding confirms that it is the protein-rich phase that gelates; see **Figure 3c** for an illustration. Having characterised the equilibrium of the mixtures belonging to different classes, we continue with the kinetics of assembly and phase separation in the next section.

## Kinetic theory of assembly at phase equilibrium

Building upon the thermodynamic framework discussed in the previous sections, we devise a non-equilibrium kinetic theory for molecular assembly at non-dilute conditions, where the interactions can give rise to coexisting phases. Here, we restrict ourselves to the case where each phase is homogeneous and at phase equilibrium but not at assembly equilibrium (**Bauermann et al., 2022**), i.e., **Equation 6a**, **Equation 6b** is fulfilled during the kinetics while **Equation 3** is not satisfied in general. This partial equilibrium holds when the molecular transitions among assemblies are slow compared to phase separation. This case is often referred to as reaction-limited (**Miangolarra et al., 2021**; **Milo and Phillips, 2015**) and applies particularly well to molecular assemblies involving biological enzymes (**Bar-Even et al., 2011**). For simplicity, we present the kinetic theory and discuss the results for two coexisting phases.

We tailor the concepts developed in **Bauermann et al., 2022** to the case of incompressible systems, $d\nu_i/dt = 0$ and $d\nu_s/dt = 0$, and volume conserving assembly kinetics, $\sum_{i=1}^{M} r_i^{\text{I/II}} = 0$, where $r_i^{\text{I/II}}$ denotes the assembly rate of assembly $i$ in each phase. In this case, the total system volume $V = V^{\text{I}} + V^{\text{II}}$ is constant, i.e., $dV/dt = 0$, and the volume fractions of the assembly of size $i$, $\phi_i^{\text{I/II}}$, is governed by:

$$\frac{d}{dt}\phi_i^{\text{I/II}} = r_i^{\text{I/II}} - j_i^{\text{I/II}} - \frac{\phi_i^{\text{I/II}}}{V^{\text{I/II}}}\frac{d}{dt}V^{\text{I/II}}, \tag{14}$$

while the solvent volume fraction in each phase is given as $\phi_s^{\text{I/II}} = 1 - \phi_{\text{tot}}^{\text{I/II}}$ with $\phi_{\text{tot}}^{\text{I/II}} = \sum_{i=1}^{M} \phi_i^{\text{I/II}}$. **Equation 14** states that the volume fraction of assemblies in each phase $\phi_i^{\text{I/II}}$ can vary due to three factors: the formation or dissolution of assemblies within the same phase (first term on the r.h.s), diffusion through the phase boundary (second term on the r.h.s), where $j_i^{\text{I/II}}$ denote the diffusive exchange rates between the phases, and changes of the respective phase volumes $V^{\text{I/II}}$ (last term on the r.h.s.). For more information, we refer the reader to Appendix 6. The kinetics of phase volumes follows

$$\frac{d}{dt}\left(\ln\frac{V^{\text{I/II}}}{V}\right) = -j_s^{\text{I/II}} - \sum_{i=1}^{M} j_i^{\text{I/II}}. \tag{15}$$

Moreover, mass conservation at the interface implies that the diffusive exchange rates of assemblies in the two phases are related via

$$j_i^{\text{I}} = -j_i^{\text{II}}\frac{V^{\text{II}}}{V^{\text{I}}} \tag{16}$$

and analogously for the solvent $j_s^{\text{I}} = -j_s^{\text{II}}V^{\text{II}}/V^{\text{I}}$. Thus, the assembly kinetics conserves the total volume fraction defined as $\phi_{\text{tot}} = \left(\phi_{\text{tot}}^{\text{I}}V^{\text{I}} + \phi_{\text{tot}}^{\text{II}}V^{\text{II}}\right)/V = 1 - \phi_s$. The exchange rates are determined by the conditions that maintain phase equilibrium, $d\mu_i^{\text{I}}/dt = d\mu_i^{\text{II}}/dt$ and $d\Pi^{\text{I}}/dt = d\Pi^{\text{II}}/dt$, where $\mu_i^{\text{I/II}}$ are the exchange chemical potentials of the monomers in an assembly of size $i$ **Equation 2**, and $\Pi^{\text{I/II}}$ are the osmotic pressures in each phase; for more information, see Appendix 6.

Using our kinetic theory, we can study the relaxation toward thermodynamic equilibrium which corresponds to simultaneous phase and assembly equilibrium. To account for association and dissociation processes associated with the reaction scheme in **Figure 1**, the phase-dependent net reaction rate for the formation of a $(i + j)$-mer starting from a $i$-mer and a $j$-mer and vice versa are set by the exchange chemical potentials via the following law of mass action **Mysels, 1956**:

$$\Delta r_{ij}^{\text{I/II}} = k_{ij}\left[\exp\left(\frac{i\mu_i + j\mu_j}{k_{\text{B}}T}\right) - \exp\left(\frac{(i+j)\mu_{i+j}}{k_{\text{B}}T}\right)\right], \tag{17}$$

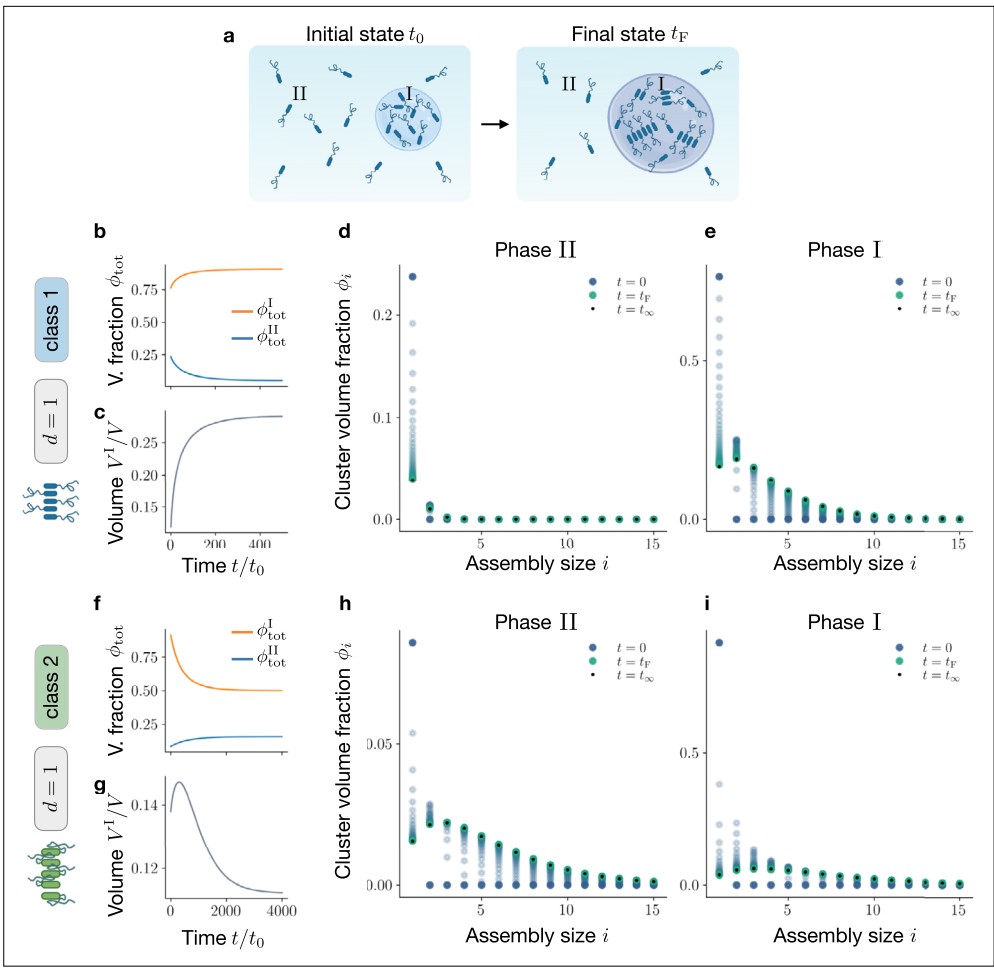

**Figure 4.** Assembly kinetics at phase equilibrium. Assuming that the relaxation to phase equilibrium is fast compared to assembly kinetics, we study the slow relaxation to assembly equilibrium in a compartmentalised system. (**a**) In the sketch, starting from an initial state composed of monomers and solvent only, assemblies selectively appear in phase I, increasing its volume $V^{\mathrm{I}}$ and total volume fraction $\phi_{\mathrm{tot}}^{\mathrm{I}}$. (**b**, **c**) For Class 1, as time proceeds, the total macromolecule volume fraction in the two phases, $\phi_{\mathrm{tot}}^{\mathrm{I/II}}$, changes inducing the growth of phase I. In (**d** and **e**). we show the time evolution of the full-size distribution in phase II and I, respectively. (**f**, **g**) For Class 2, as time proceeds, changes in total macromolecule volume fraction in the two phases cause a shrinkage of phase I. This is reminiscent of recent experimental findings that quantify droplet volume changes along with droplet ageing (*Jawerth et al., 2020*). (**h**, **i**) Time evolution of assembly volume fractions $\phi_i(t)$ in phase II and I, respectively. Time is measured in units of the discretisation time step $t_0 = 0.01/\tilde{k}$, where the rate $\tilde{k}$ is introduced in *Equation 40*.

where $k_{ij}$ is a size-dependent kinetic rate coefficient, see Appendix 6 for details. The assembly rates $r_i^{\mathrm{I/II}}$ entering *Equation 14* can finally be expressed as a function of the $(i+j)$-mer exchange rate $\Delta r_{ij}^{\mathrm{I/II}}$, by

$$r_i^{\mathrm{I/II}} = \frac{i}{2} \sum_{j,k:j+k=i} \Delta r_{jk}^{\mathrm{I/II}} - i \sum_{j} \Delta r_{ij}^{\mathrm{I/II}}. \tag{18}$$

In the next section, we compare the kinetics of systems belonging to Class 1 and 2, for $d = 1$.

## Assembly kinetics in coexisting phases

By integrating *Equation 14* numerically, we obtain the time evolution of $\phi_i^{\mathrm{I/II}}(t)$ and $V^{\mathrm{I}}(t)$, provided their initial values at $t = 0$, $V^{\mathrm{I}}(t = 0)$, and $\phi_i^{\mathrm{I/II}}(t = 0)$, at phase equilibrium. Specifically, we consider an initial state solely composed of solvent and monomers demixed into a monomer-rich and a monomer-poor phase (labelled with I and II, respectively, see the illustration in *Figure 4a*). For simplicity, we focus on

linear assemblies ($d = 1$) and highlight differences between Class 1 and 2; for parameters, see caption of *Figure 4*. We note that the kinetics for $d > 1$, where gelation can occur, would require removing the upper bound in assembly size, i.e., studying trajectories in an $M$-dimensional space, where $M \to \infty$ in the thermodynamic limit. This case is numerically challenging and we leave its investigation for future work.

For Class 1, as monomers start forming assemblies, the mixing entropy decreases. As a result, the total amount of protein in the monomer-rich phase, $\phi_{\mathrm{tot}}^{\mathrm{I}}$, increases while $\phi_{\mathrm{tot}}^{\mathrm{II}}$ decreases (*Figure 4b*). Such changes in total protein volume fractions induce phase volume variations (*Figure 4c*). In particular, remaining within Class 1, since the monomer enrichment of phase I is less pronounced than the monomer depletion of phase II, the volume of the protein-rich phase $V^{\mathrm{I}}$ increases. An important finding of our work is that the distribution of assembly size evolves differently in each phase (*Figure 4d and e*; and SI *Appendix 1—Video 1*). In phase II, which is initially poor in monomers, assemblies grow slowly toward an equilibrium distribution where the volume fractions monotonously decrease with assembly size, following an exponential decay. The kinetics in the initially monomer-rich phase I is fundamentally different. First, a very pronounced peak of intermediate-sized assemblies develops quickly. The faster kinetics compared to phase II is caused by monomer diffusion from II to I, which leads to negative feedback for assembly in II and positive feedback in I. This observation is reminiscent of studies on dilute, irreversible aggregation in coexisting phases (*Weber et al., 2019a*). The most abundant populations of intermediate-sized assemblies shrink slowly in time feeding the growth of larger assemblies. The resulting equilibrium distribution shows a notable peak of intermediate-sized assemblies followed by an exponential decay. Thus, the difference in the kinetics between the phases is dominantly a consequence of the fact that each phase strives towards a significantly different equilibrium distribution.

Assemblies belonging to Class 2 exhibit a different behaviour. In this class, as monomers assemble, their interaction propensity decreases. As a result, depending on the values of $\chi$ and $\chi'$, the total amount of protein in the protein-rich phase, $\phi_{\mathrm{tot}}^{\mathrm{I}}$, can decrease, as in the case of *Figure 4f*. For this choice of parameters, the total amount of protein in the protein-poor phase, $\phi_{\mathrm{tot}}^{\mathrm{II}}$, increases; see again

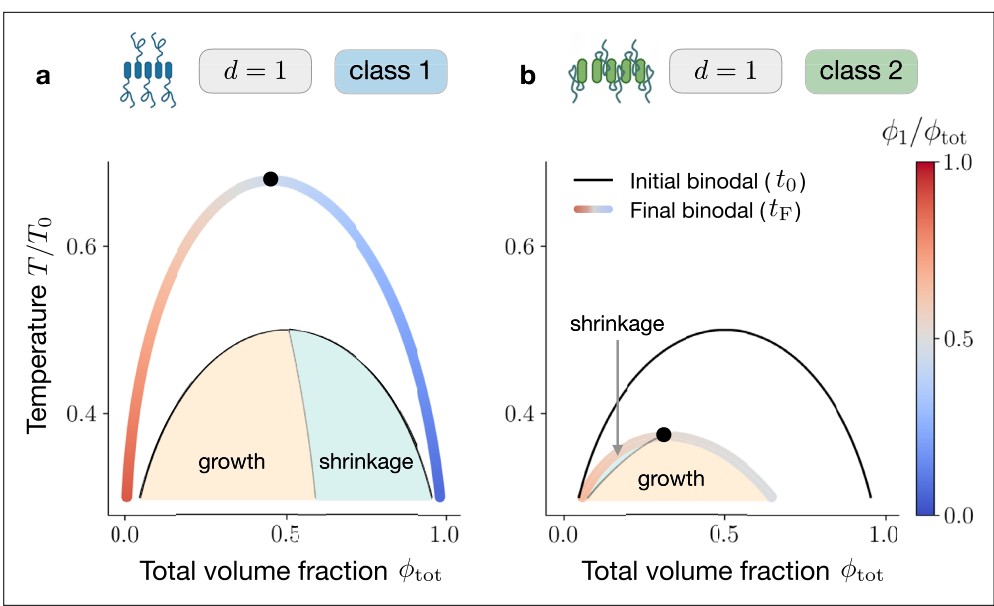

**Figure 5.** Identification of shrinkage and growth regions for different classes. Here, we study phase-separating systems initially composed of monomers only and we monitor phase volume changes as they relax to thermodynamic equilibrium. (**a**) For linear assemblies (d=1) belonging to class 1, the final binodal line (coloured curve) is wider than the initial one (black curve), corresponding to monomers and solvent only (black curve). Areas in orange and light blue correspond to growth and shrinkage of the $\phi_{\mathrm{tot}}$-rich phase (phase I), respectively. (**b**) The behaviour of linear assemblies (d=1) belonging to class 2 is remarkably different. Since, in this class, the interaction with the solvent is screened, the final binodal is shrunk compared to the initial one. As a consequence of the shrinkage, the domain corresponding to phase I growth (light blue area) precedes in $\phi_{\mathrm{tot}}$ the shrinkage domain (orange area), for class 2.

*Figure 4f*. Furthermore, in this case, the initial rapid decrease in $\phi_{\text{tot}}^{\text{I}}$, followed by more moderate changes at later times, induces a non-monotonic phase volume variation (*Figure 4g*), leading to a net shrinkage of phase I. Further investigation is needed to shed light on the physical mechanisms underlying this non-monotonic volume variation. In *Figure 4h, i* and SI *Appendix 1—Video 2*, we show how the volume fractions $\phi_i(t)^{\text{I/II}}$ for each assembly size $i$ evolve in both phases I and II. In the next section, we explore under which conditions phases grow or shrink during the relaxation to equilibrium.

## Assembly formation can increase or decrease condensate volume

Here, we discuss changes in phase volumes caused by the assembly kinetics introduced in Sec. Kinetic theory of assembly at phase equilibrium. In particular, we focus on mixtures initially demixed in two phases, both composed of only monomers, and let the system relax to thermodynamic equilibrium. We then assess for which values of the control parameters $\phi_{\text{tot}}$ and $T$, the formation of assemblies in both phases leads to a growth of the $\phi_{\text{tot}}$-rich phase (phase I) and vice versa. Moreover, we distinguish the two protein classes introduced in Sec. Scaling of molecular volumes, internal free energies and interaction energies with assembly size.

To this end, we compare the phase diagram corresponding to the initial system, composed of monomers only, with the equilibrium phase diagram in which large assemblies populate the mixture. In *Figure 5a*, we show the initial and final equilibrium binodals (black and coloured curve, respectively), for the case of linear assemblies ($d = 1$) belonging to class 1. In this case, the domain corresponding to demixing enlarges once the system reaches its equilibrium state, i.e., assembly facilitates phase separation. We focus on the $\phi_{\text{tot}} - T$ domain enclosed by the black curve, where the system is phase separated at all times and compute the initial and final $\phi_{\text{tot}}$-rich phase volumes via the total volume fraction conservation $V^I(t)/V = (\phi_{\text{tot}} - \phi_{\text{tot}}^{\text{II}}(t))/(\phi_{\text{tot}}^{\text{I}}(t) - \phi_{\text{tot}}^{\text{II}}(t))$. As displayed in *Figure 5a*, this allows us to identify two parameter regimes: at low $\phi_{\text{tot}}$ (orange area), the protein-rich phase grows as assemblies form, while above the dashed grey line (light blue area), it shrinks. Remarkably, linear assemblies ($d = 1$) belonging to class 2 exhibit a completely different behaviour, see *Figure 5b*. In this case, assembly formation shrinks the domain corresponding to demixing, thereby suppressing phase separation. In the domain enclosing the coloured curve, we can compute the initial and final volumes of the protein-rich phase for each value of $\phi_{\text{tot}}$ and $T$. In contrast to the previous case, we find that at low $\phi_{\text{tot}}$ (light blue area), the protein-rich phase shrinks as assemblies are formed, while for higher $\phi_{\text{tot}}$ values (orange area) condensate volume grows, as illustrated in *Figure 5b*.

## Discussion

We discuss an extension of the classical theory of molecular assembly (*Stockmayer, 1943*; *Krapivsky et al., 2010*; *Israelachvili, 2015*) to non-dilute conditions and study it for cases where assemblies can phase-separate from the solvent and gelate. This extension relies on a thermodynamic free energy governing the interactions among all assemblies of different sizes and the solvent. We propose two classes to account for protein interactions relevant to biological systems that can phase separate and form assemblies. Classes differ in the way how energetic parameters for interactions and internal free energies depend on assembly size.

Using our theory, we report several key findings that arise from non-dilute conditions and the ability of assemblies to form a condensed phase. First, size distributions, in general, differ between the phases. In particular, monomers are not necessarily the most abundant species, and distribution tails can significantly deviate from the exponential decay known for classical assembly at dilute conditions (*Krapivsky et al., 2010*). Interestingly, this statement also applies to conditions below the saturation volume fraction beyond which phase separation can occur. Second, we showed that by lowering the temperature, the protein-rich phase can gelate, i.e., it consists of a single connected assembly of volume equal to the protein-rich phase (a gel). Upon gelation, the composition of the protein-poor phase changes continuously, while the protein-rich liquid phase discontinuously transits to the gel phase. Third, when monomers start assembling in the respective phases, the volume of the protein-rich phase can grow or shrink depending on the molecular interactions among the constituents.

Our key findings are consistent with recent experimental observations in living cells and in vitro assays using purified proteins. A decrease in droplet volume has been observed in phase-separated condensates composed of purified FUS proteins (*Jawerth et al., 2020*). Up to now, it has remained

unclear whether this kinetics relies on a glass transition as suggested in the discussion of *Jawerth et al., 2020*, or on the formation of FUS oligomers in the protein-rich phase. However, a potential hint comes from independent studies, which indicate that FUS can form amyloid-like assemblies that are associated with neurodegenerative disorders (*Zbinden et al., 2020*), at similar conditions (*Patel et al., 2015*; *Kato et al., 2012*). Moreover, the gelation of dense protein condensates upon temperature and heat stress was suggested in several in vivo studies in living cells (*Franzmann et al., 2018*). The transition to a gelated condensate is believed to provide a protection mechanism for the protein expression machinery in the case of intracellular stress. Recently, in vitro experiments using purified proteins indicate anomalous size distributions of phase-separating proteins below saturation (*Kar et al., 2022*). Our theoretically predicted size distributions could be compared to systematic experimental studies using single molecule techniques such as FRET. From this comparison, protein interactions of assembly-prone and phase-separating proteins can be characterised using our proposed classes.

Though many biologically relevant assembly processes are reversible and governed by thermodynamic principles, there are also a large number of assemblies that are persistently maintained away from equilibrium. For example, the formation or disassembly of assemblies can depend on the hydrolysis of ATP (*Hartl et al., 2011*) while it can also act as cosolute (*Patel et al., 2017*; *Hautke and Ebbinghaus, 2023*). Since fuel levels are approximately kept constant in living cells, fuel-driven assembly processes are maintained away from equilibrium and thus cannot relax to thermodynamic equilibrium. It is an exciting extension of our work to consider fuel and waste components and how distributions of assembly sizes and the gelation of condensates are affected when maintained away from equilibrium.

## Acknowledgements

We thank J Bauermann, K Alameh, P McCall, T Harmon, L Hubatsch, L Jawerth, and F Jülicher for fruitful discussions about the topic. We thank C Seidel and T Franzmann for pointing out the relevance of your theory for protein aggregation in biomolecular condensates. We acknowledge J-F Joanny for pointing out the references *Blankschtein et al., 1985*; *Tanaka, 2011*. We thank S Safran for the very insightful feedback on the manuscript. We thank J Bauermann, S Horvát, and C Duclut for help improving the Mathematica code. Bartolucci and C Weber acknowledge the SPP 2191 'Molecular Mechanisms of Functional Phase Separation' of the German Science Foundation for financial support. C Weber acknowledges the European Research Council (ERC) under the European Union's Horizon 2020 research and innovation programme (Fuelled Life, Grant Number 949021) for financial support. Figures created with BioRender.com.

## Additional information

### Funding

| Funder | Grant reference number | Author |
| --- | --- | --- |
| Deutsche Physikalische Gesellschaft | SPP2191 | Giacomo Bartolucci Christoph A Weber |
| European Commission | 10.3030/949021 | Christoph A Weber |

The funders had no role in study design, data collection and interpretation, or the decision to submit the work for publication.

### Author contributions

Giacomo Bartolucci, Conceptualization, Formal analysis, Investigation, Visualization, Writing – original draft, Writing – review and editing; Ivar S Haugerud, Investigation, Writing – original draft, Writing – review and editing; Thomas CT Michaels, Conceptualization, Formal analysis, Supervision, Investigation, Writing – original draft, Writing – review and editing; Christoph A Weber, Conceptualization, Formal analysis, Supervision, Funding acquisition, Investigation, Writing – original draft, Writing – review and editing

**Author ORCIDs**
Giacomo Bartolucci http://orcid.org/0000-0002-3128-1154
Christoph A Weber https://orcid.org/0000-0001-6279-0405

Reviewer #3 (Public review): https://doi.org/10.7554/eLife.93003.3.sa1
Author response https://doi.org/10.7554/eLife.93003.3.sa2

**Data availability**
Code is available on GitHub at https://github.com/GiacoBarto/AssemblyAndPhaseSeparation (copy archived at *Bartolucci, 2026*).

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

# Appendix 1

## Parameters used and movies

Parameters used in each figure is shown in *Appendix 1—table 1*.

**Appendix 1—table 1.** Parameters corresponding to the figures in the main text.
We made use of the temperature scale $T_0 = \chi/k_B$.

| Figure | $e_{\text{int}}/\chi$ | $s_{\text{int}}/k_B$ | $\chi'/\chi$ | $M$ | $\phi_{\text{tot}}$ | $T/T_0$ |
|---|---|---|---|---|---|---|
| *Figure 2a* | −5 | −2 | - | ∞ | - | - |
| *Figure 2b* | −0.5 | −2 | - | ∞ | - | - |
| *Figure 2c* | - | −2 | - | ∞ | - | 0.6 |
| *Figure 2d* | - | −2 | 0.2 | ∞ | - | 0.25 |
| *Figure 3d* | −2 | −2 | - | ∞ | - | - |
| *Figure 4b-e* | −1 | −2 | - | 15 | 0.3 | 0.45 |
| *Figure 4f-i* | −1 | −2 | 0.2 | 15 | 0.2 | 0.35 |
| *Figure 5a* | −1 | −2 | - | 15 | - | - |
| *Figure 5b* | −1 | −2 | 0.2 | 15 | - | - |

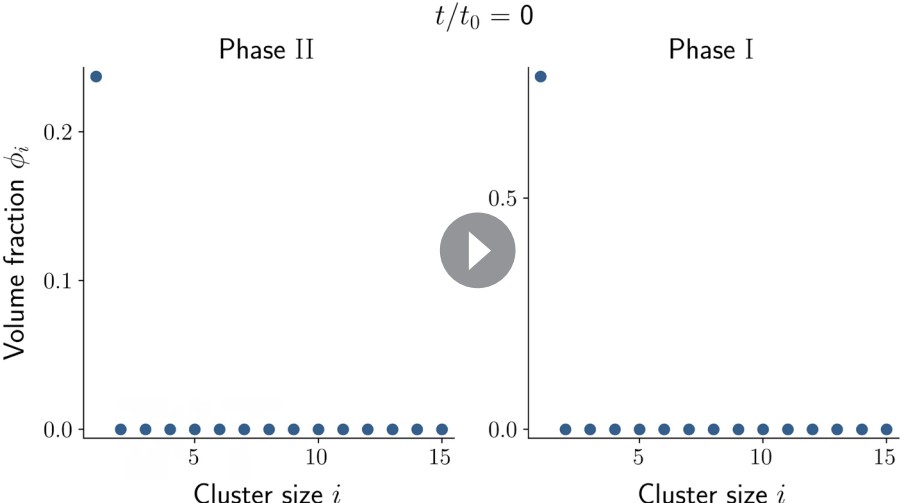

**Appendix 1—video 1.** Shows the time evolution of the assembly volume fractions in both phases, for Class 1.
Parameters are the same as *Figure 4b-e*.
https://elifesciences.org/articles/93003/figures#video1

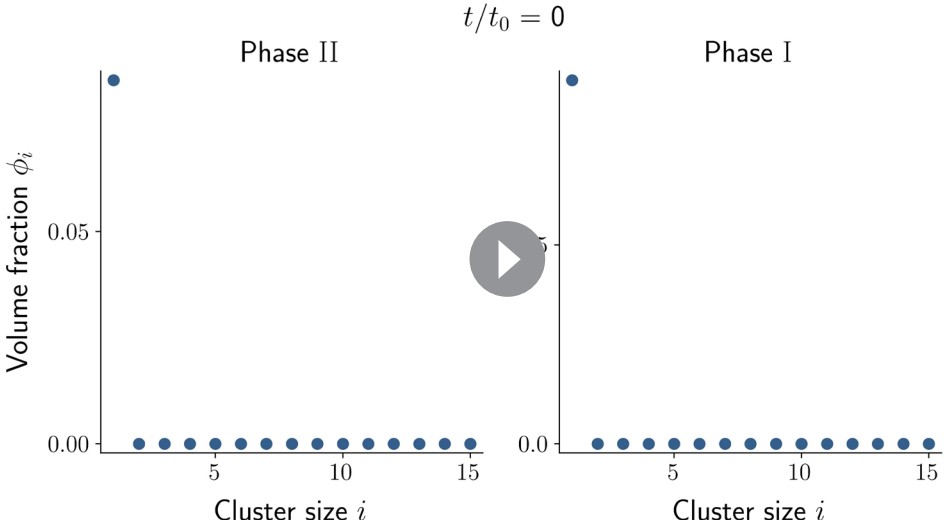

**Appendix 1—video 2.** Shows the time evolution of the assembly volume fractions in both phases for Class 2. Parameters are the same as *Figure 4f-i*.

https://elifesciences.org/articles/93003/figures#video2

## Appendix 2

## Scaling laws for internal and interaction energies

Here, we provide a physical interpretation of the internal free energy $\omega_i$. For simplicity, we consider a homogeneous system solely composed of assemblies of size $i$, characterised by the volume fraction vector $\phi^{(i)}$, with $\phi_i^{(i)} = 1$ and $\phi_j^{(i)} = 0$ for $i \neq j$. Making use of *Equation 1*, the internal free energy of such systems can be written as

$$\omega_i = f(\phi^{(i)})\nu_1 - k_B T\, i^{-1} \ln i^{-1}, \tag{19}$$

with $f(\phi^{(i)})\nu_1$ being the free energy associated with each monomer belonging to the $i$-th assembly. The second term in the equation above is the conformational entropy that stems from having more accessible states with increasing assembly size. Thus, *Equation 19* allows interpreting $\omega_i$ as the free energy of monomers inside an assembly of size $i$, coming only from bonds between monomers. To quantify it, we introduce the number of binding sites for each monomer, $z$. Following *Israelachvili, 2015*, we distinguish between $n_b$ monomers at the boundaries of the assembly, and $(i - n_b)$ in the assembly bulk. Monomers in the bulk can saturate all their $z$ binding sites, while, in general, monomers at the boundaries are able to saturate only $z_b < z$. Thus, we get

$$\begin{aligned} \omega_i &= \frac{z(i - n_b) + z_b n_b}{2i}\,\Delta\omega \\ &= \frac{z}{2}\,\Delta\omega - (z - z_b)\frac{n_b}{2i}\,\Delta\omega, \end{aligned} \tag{20}$$

where $\Delta\omega$ is the free energy associated with the formation of a single bond, that is composed of an energetic and an entropic part. The factor two avoids double-counting.

We describe three species of assemblies: linear, disc-like, and three-dimensional. These can be realised by varying the number of binding sites and their orientation (*Nandi et al., 2022*). Linear assemblies ($d = 1$) are defined to have only two binding sites. They can be pictured as one-dimensional semi-flexible assemblies with no loops, leading to $n_b = 2$, $z = 2$ and $z_b = 1$. Planar assemblies ($d = 2$) are defined to have $z > 2$ co-planar binding sites, for which $n_b \simeq \sqrt{i}$. Three-dimensional assemblies ($d = 3$) are characterised by $z > 2$ binding sites with no precise orientation leading to $n_b \simeq i^{\frac{2}{3}}$. Summing up, we get

$$n_b \simeq i^{\frac{d-1}{d}}, \tag{21}$$

that inserted in *Equation 20*, decomposing $\Delta\omega$ in its energetic and entropic contribution, gives

$$\omega_i = \alpha\left(e_{int} - Ts_{int}\right) - \left(\frac{e_{int} - Ts_{int}}{i^{1/d}}\right), \tag{22}$$

where $\alpha$ is a constant that depends on number and geometry of the binding sites. Identifying $\omega_\infty = \alpha\left(e_{int} - Ts_{int}\right)$, *Equation 22* leads to *Equation 8*, in the main text. The constant terms $\omega_\infty$, does not affect chemical nor phase equilibrium. However, in the case of $d = 2, 3$ and $M \to \infty$, $\omega_\infty$ it becomes important to study the gelation of the protein-rich phase, see Appendix 4. In *Equation 8*, the second term represents a boundary interaction penalty, accounting for the fact that monomers at the assembly boundary can realise fewer internal bonds than monomers at the assembly bulk, in analogy with the physical origin of surface tension.

We now discuss the size dependence of the interaction parameters $\chi_{ij}$. Starting from a lattice model, these parameters can be expressed in terms of the energetic parameters $e_{ij}$ corresponding to having two neighbouring monomers belonging to $i$ and $j$. In particular, $\chi_{ij} = 2e_{ij} - e_{ii} - e_{jj}$. Assuming that the energies associated with monomer-monomer interactions do not vary within assemblies, i.e., $e_{ij} = e_{11}$ is constant, we get $\chi_{ij} = 0$. Moreover, we now discuss the scaling of $\chi_{is} = 2e_{is} - e_{ii} - e_{ss}$. If the monomer-solvent interactions are also chosen to be size-independent, i.e., $e_{is} = e_{1s}$, we get $\chi_{is} = 2e_{1s} - e_{11} - e_{ss} = \chi$. This explains the scaling in Class 1 (see *Equation 9*).

However, many proteins of interest screen their hydrophobic interaction when forming assemblies (*Hughes et al., 2018*; *Luo et al., 2018*; *Seim et al., 2022*), implying that the interactions between

monomers in assembly $i$ with solvent (s) $e_{is}$ varies with assembly size $i$. In each assembly, this energy per monomer comes from two contributions. The first corresponds to monomers in the bulk which are $(n - n_b)$ and have interaction with solvent $e'_{1s}$. The second one corresponds to the $n_b$ monomers at the assembly boundary, characterised by interaction with solvent $e_{1s}$. We get

$$e_{is} = \frac{e'_{1s}(i - n_b) + e_{1s}n_b}{i}. \tag{23}$$

Using the scaling of $n_b/i$ already introduced above in the discussion of the internal free energy scaling, *Equation 21*, we obtain *Equation 10*. By abbreviating $\chi'_{is} = 2e'_{1s} - e_{11} - e_{ss} = \chi'$; this case corresponds to Class 2.

## Appendix 3

## Linear assemblies belonging to class 1

For class 1 and $d = 1$, *Equation 4* reads

$$\phi_i = i \left( \frac{\phi_1}{\tilde{\phi}} \right)^i \tilde{\phi},$$

(24)

where we have introduced the characteristic volume fraction

$$\tilde{\phi} = \exp \left( \frac{e_{\text{int}} - s_{\text{int}} T}{k_B T} - 1 \right).$$

(25)

It is straightforward to verify that the latter volume fraction is proportional to the assembly threshold defined in *Equation 11* i.e., $\tilde{\phi} = e/(1 - e)^2 \phi^* \sim \phi^*$.

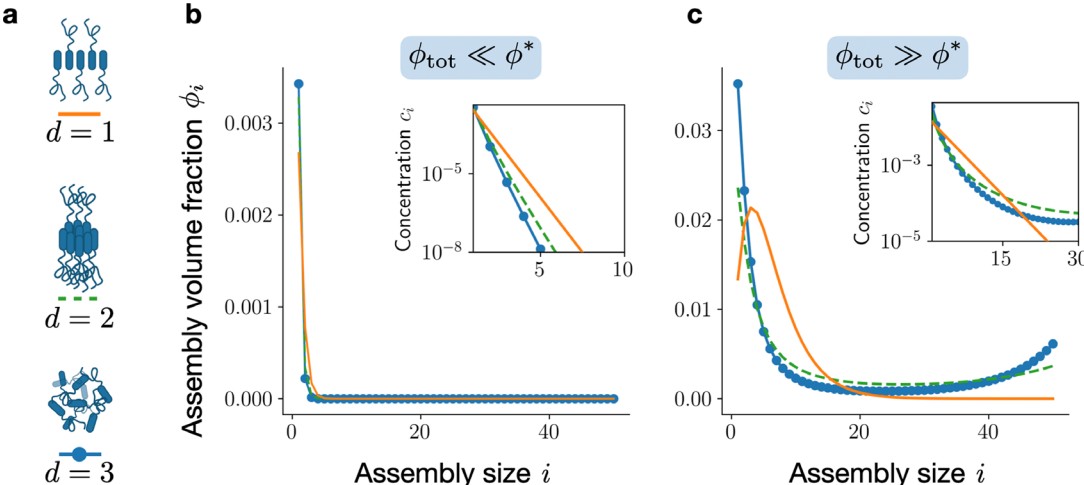

**Appendix 3—figure 1.** A volume fraction threshold separates two assembly regimes in homogeneous systems. (**a**) Illustration of assemblies belonging to Class 1 with different spatial dimensions. (**b**) Assembly size distribution at low total macromolecular volume fraction: $\phi_{\text{tot}} = 0.2\phi^*$. Disregarding assembly dimension, $d$, the macromolecules are mainly in the monomer state, i.e., $\phi_1 \simeq \phi_{\text{tot}}$. (**c**) For $\phi_{\text{tot}} = 10\phi^*$, the monomer volume fraction saturates at $\phi_1 \simeq \phi^*$ and big assemblies begin to populate the system. For linear assemblies (corresponding to $d = 1$ in *Equation 8*), the distribution becomes peaked at an intermediate value $i_{\max} > 1$ and then exponentially cut off. For planar and three-dimensional assemblies, $d = 2, 3$, the distribution becomes bimodal, with peaks at $i = 1$ and $i = M$, the maximum assembly size ($M = 50$). This bimodal behaviour hints at the emergence of a gelation transition in the limit $M \to \infty$. In the insets, we show the scaling of concentrations $c_i = \phi_i/\nu_i$ with assembly size. For $d = 2, 3$, and above the $\phi^*$ threshold, deviations from the classical exponential decay are present. Here $e_{\text{int}}/\chi = -1$, $s_{\text{int}}/k_B = -1$, $M = 50$, $T/T_0 = 0.25$.

In *Appendix 3—figure 1*, we show the assembly size distribution in homogeneous mixtures obtained by numerically solving *Equation 4* together with *Equation 5*, with a cut-off $M = 50$. We characterise the behaviour of assemblies with different spatial dimensions $d = 1, 2, 3$, see *Appendix 3—figure 1a*. For dilute solutions, corresponding to $\phi_{\text{tot}} \ll \phi^*$, the size distribution is dominated by monomers while larger assemblies have vanishing volume fraction, i.e., $\phi_1 \simeq \phi_{\text{tot}}$, see *Appendix 3—figure 1b*. For $\phi_{\text{tot}} \gg \phi^*$, the monomer volume fraction saturates at $\phi_1 \simeq \phi^*$ and assemblies begin to populate the system. As depicted in *Appendix 3—figure 1b*, above this threshold, the size distribution depends crucially on assembly dimension $d$. For linear assemblies ($d = 1$ in *Equation 8*), the distribution becomes peaked at a value $M > 1$ and then exponentially decays. For planar and three-dimensional assemblies, $d = 2, 3$ in *Equation 8*, the distribution becomes bimodal peaked at $i = 1$ and $i = M$, the maximum assembly size ($M = 50$ in *Appendix 3—figure 1c*). The behaviour of the system at high density can be quantitatively studied by performing the thermodynamic limit, i.e., $M \to \infty$. Within this limit, the series defined in the conservation law, *Equation 5* can be explicitly solved, leading to

$$\frac{\phi_1}{\tilde{\phi}} = \frac{1 + 2\frac{\phi_{\text{tot}}}{\tilde{\phi}} - \sqrt{1 + 4\frac{\phi_{\text{tot}}}{\tilde{\phi}}}}{2\frac{\phi_{\text{tot}}}{\tilde{\phi}}}. \tag{26}$$

Recalling that $\tilde{\phi} = e/(1-e)^2 \phi^*$, this leads to $\phi_1 \simeq \phi_{\text{tot}}$, in the regime $\phi_{\text{tot}} \ll \phi^*$, while for $\phi_{\text{tot}} \gg \phi^*$, we get $\phi_1 \simeq \phi^*$.

The maximum of the volume fraction distribution in *Equation 24* can be obtained imposing $\partial_i \phi_i = 0$, leading to

$$i_{\text{max}} = \frac{1}{\ln(\tilde{\phi}/\phi_1)} \simeq \sqrt{\frac{\phi_{\text{tot}}}{\tilde{\phi}}}. \tag{27}$$

The approximate expression on the right hand is obtained using *Equation 26* and expanding for $\phi_{\text{tot}}/\tilde{\phi} \gg 1$.

The average $\langle i \rangle = \sum i\phi_i / \sum \phi_i$ is given by

$$\langle i \rangle = \frac{\phi_1}{\phi_{\text{tot}}} \frac{3 - \phi_1/\tilde{\phi}}{(1 - \phi_1/\tilde{\phi})^3} \simeq 2\, i_{\text{max}}, \tag{28}$$

where we expanded for $\phi_{\text{tot}}/\tilde{\phi} \gg 1$ to obtain the approximate expression.

We can also derive an explicit expression for the free energy as a function of the conserved quantity $\phi_{\text{tot}}$. This is achieved by plugging $\phi_i$ given in *Equation 24* into *Equation 1* Omitting linear terms, which do not influence phase equilibrium, we get

$$f_{\text{sol}} = \frac{k_B T}{\nu_1} \left[ (1 - \phi_{\text{tot}}) \ln(1 - \phi_{\text{tot}}) + \phi_{\text{tot}} \ln \frac{\phi_1}{\tilde{\phi}} - \frac{\phi_1}{1 - \phi_1/\tilde{\phi}} + \frac{\chi}{k_B T} \phi_{\text{tot}}(1 - \phi_{\text{tot}}) \right], \tag{29}$$

where the dependence of $\phi_1$ on $\phi_{\text{tot}}$ is expressed in *Equation 26*.

## Appendix 4

## Gelation transition for two- and three-dimensional assemblies

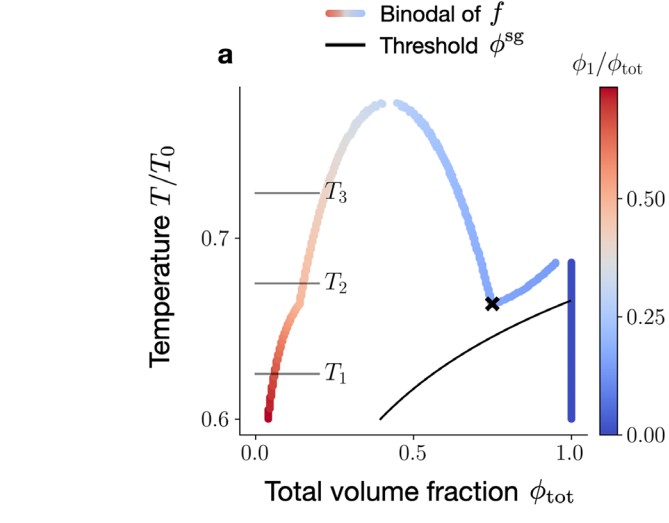

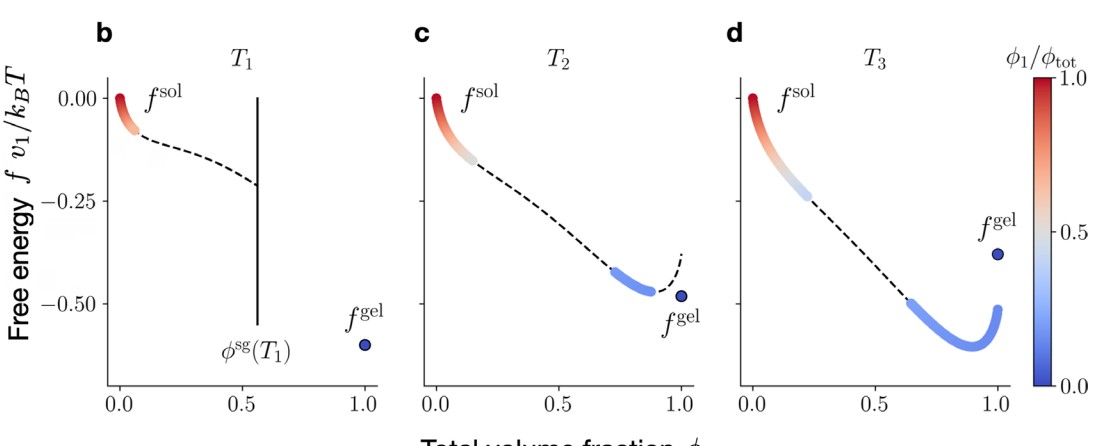

**Appendix 4—figure 1.** Gel-sol free energies. (**a**) The coloured curve indicates the binodal obtained with the Maxwell construction for $f = f_{\text{sol}} + f_{\text{gel}}$, together with the estimate $\phi^{\text{sg}}(T)$ (black line, defined in *Equation 31*) for the transition between homogeneous and gel states. The black cross 'x' indicates the triple point. (**c-e**) Maxwell construction for three different temperature values, the coloured and black, dashed curves represent convex and concave branches of $f$, respectively. Parameters are the same as those in *Figure 3*; see *Appendix 1—table 1*.

As outlined in *Appendix 3—figure 1*, for $d = 2, 3$, at high $\phi_{\text{tot}}$ for $M$ finite, the size distribution shows a bimodal behaviour. This suggests a limit $M \to \infty$ that the system undergoes a gelation transition, which is defined as the emergence of an assembly that is comparable with the system size (*Tanaka, 2011*; *Israelachvili, 2015*; *Deviri and Safran, 2020*). To estimate the $\phi_{\text{tot}}$ value at which the transition occurs, we recall *Equation 4* and consider the series

$$\sum_{i=1}^{\infty} \phi_i = \sum_{i=1}^{\infty} i \left( \frac{\phi_1}{\tilde{\phi}} \right)^i \exp \left( \frac{\Delta\omega}{k_{\text{B}}T} i^{\frac{d-1}{d}} - 1 \right). \tag{30}$$

We note that when $N \to \infty$, this series converges only if $\phi_1/\tilde{\phi} \leq 1$. Thus, we get an upper bound for the series, namely

$$\sum_{i=1}^{\infty} \phi_i \leq \sum_{i=1}^{\infty} i \exp \left( \frac{\Delta\omega}{k_{\text{B}}T} i^{\frac{d-1}{d}} - 1 \right) \equiv \phi^{\text{sg}}. \tag{31}$$

Approximating the series with the integral, we get an estimation for $\phi^{\mathrm{sg}}$:

$$\phi^{\mathrm{sg}} = \begin{cases} 2\dfrac{\left(6 - 6\Delta\omega + 3\Delta\omega^2 - \Delta\omega^3\right)}{\Delta\omega^4}\tilde{\phi}, & d = 2\,, \\[2ex] -\dfrac{3}{2}\dfrac{\Delta\omega^3}{\left(2 - 2\Delta\omega + \Delta\omega^2\right)}\tilde{\phi}, & d = 3\,. \end{cases} \tag{32}$$

By the Maxwell construction, *Equation 7a*, *Equation 7b* with the free energy *Equation 12*, we can study the interplay between the gelation transition and phase separation. Here, the parameter $\omega_\infty$ plays a crucial role. As discussed in Appendix 2, $\omega_\infty$ contains an energetic and an entropic part and is proportional to $e_{\mathrm{int}} - Ts_{\mathrm{int}}$, the coefficient depending on assembly dimension, and number and geometry of the binding sites. Here, for simplicity, we set

$$\omega_\infty = \frac{1}{2}\left(e_{\mathrm{int}} - Ts_{\mathrm{int}}\right). \tag{33}$$

In *Appendix 4—figure 1a*, we display the result of the construction (coloured curve), where the colour code depicts the monomer fraction $\phi_1/\phi_{\mathrm{tot}}$ in the coexisting phases. Note that the branch of the binodal between the triple point (indicated with a cross) and $\phi_{\mathrm{tot}} = 1$, has the same trend as the curve that stems from the estimate $\phi^{\mathrm{sg}}$ introduced in *Equation 32* (black curve). This is expected since both curves correspond to the boundary between homogeneous mixtures and the gel state.

We also display the free energy for three temperature values corresponding to sol-gel coexistence (*Appendix 4—figure 1b*), sol-sol and sol-gel coexistence (*Appendix 4—figure 1c*), and sol-sol coexistence only (*Appendix 4—figure 1d*). The dashed lines represent values where $f$ is not convex. Notice that, for consistency, we use values of $f_{\mathrm{sol}}$ only up to $\phi^{\mathrm{sg}}$ (denoted by a vertical black line). This is because, as depicted in *Appendix 3—figure 1*, after this value, a peak at $M$, the finite cut-off used for the numerics, will appear.

## Appendix 5

### Mutual feedback between phase separation and assembly equilibria

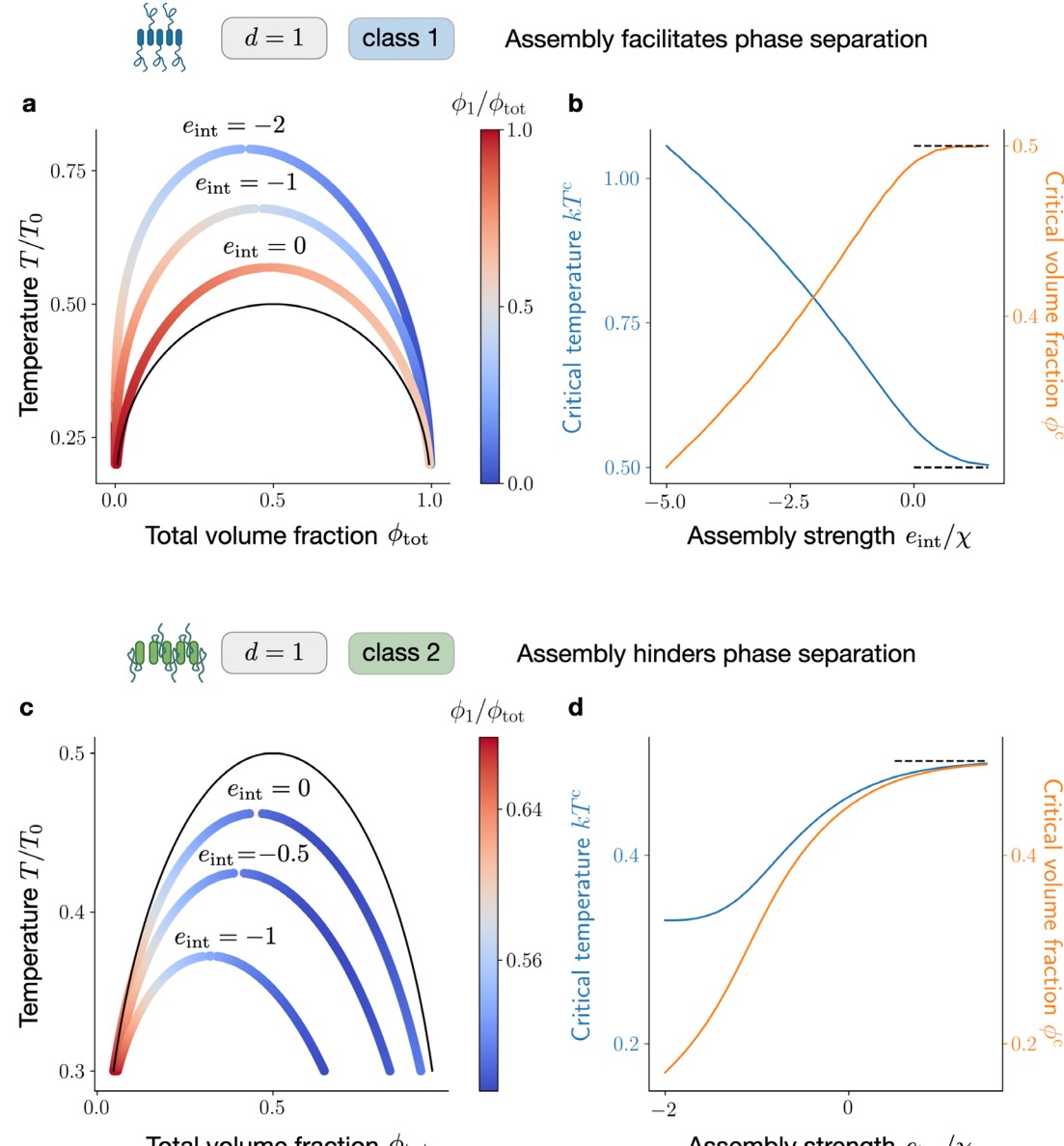

**Appendix 5—figure 1.** The influence of assemblies on the system phase behaviour. (**a**) Focusing on systems with $d = 1$ belonging to class 1, we compare three binodals corresponding to assembly strength $e_{int}/\chi = 0.5, -1, -2$ (coloured curves) and the reference binary mixture composed of monomers and solvent only (black curve). The latter can be associated with the limit $e_{int}/\chi \to \infty$. The region enclosed by the binodal, corresponding to phase separation, expands even for assemblies with no assembly energy $e_{int}/\chi = 0$. This can be explained by the entropic advantage caused by size polydispersity. (**b**) Dependence of the critical volume fraction and critical temperature on the assembly strength $e_{int}/\chi$. The presence of assemblies causes $T^c$ and $\phi^c$ to deviate from the reference values (black dashed lines) corresponding to a binary mixture with monomers and solvent only ($e_{int}/\chi \to \infty$). In particular, for Class 1, making assemblies more energetically favourable, i.e., decreasing $e_{int}/\chi$, induces an increase in $T^c$ and a decrease in $\phi^c$, in turn making phase separation more accessible. Here $s_{int}/k_B = -2$, $M = \infty$. (**c**) Comparison between three binodal lines corresponding to systems belonging to class 2 and $d = 1$, with assembly energies $e_{int}/\chi = 0, -0.5, -1$ (coloured curves) and the reference binary mixture composed of monomers and solvent only (black curve). (**d**) For Class 2, decreasing $e_{int}/\chi$, causes $T^c$ and $\phi^c$ to decrease, overall hindering phase separation. This is caused by the interaction propensity screening in monomers at the bulk of assemblies belonging to class 2; see *Equation 10*. Here $s_{int}/k_B = -2$, $M = \infty$

We first discuss how assemblies can shape the phase diagram. For linear assemblies ($d = 1$) belonging to Class 1, assemblies facilitate phase separation. Indeed, as illustrated in *Appendix 5— figure 1a-b*, increasing the relative assembly strength, i.e., decreasing $e_{int}/\chi$, leads to an upshift in critical temperature and a downshift in critical volume fraction. This trend can be explained by considering that assembly formation, even if energetically disfavoured, reduces the mixing entropy (see the first term in *Equation 1*). In *Appendix 5—figure 1a*, we show the binodal lines corresponding to three representative values of the assembly strength: $e_{int}/\chi = 0, -1, -2$. We compare them to the black curve, which corresponds to a binary mixture made of monomers and solvent only (black curve). This reference case can be thought of as the limiting case in which assemblies have an infinite energy penalty, i.e, $e_{int}/\chi = \to \infty$. In *Appendix 5—figure 1b*, we quantify the changes in critical temperature and critical volume fraction as a function of the relative assembly strength $e_{int}/\chi$. In *Appendix 5—figure 1c-d*, we illustrate the behaviour of linear assemblies ($d = 1$) belonging to Class 2. In contrast to Class 2, assemblies can suppress phase separation. Indeed, making assemblies more favourable by decreasing $e_{int}/\chi$, the critical temperature decreases, and even if the critical density decreases and the binodal shrinks, see *Appendix 5—figure 1c*. In *Appendix 5—figure 1d*, we display critical temperatures and critical volume fraction variations as a function of the relative assembly strength $e_{int}/\chi$.

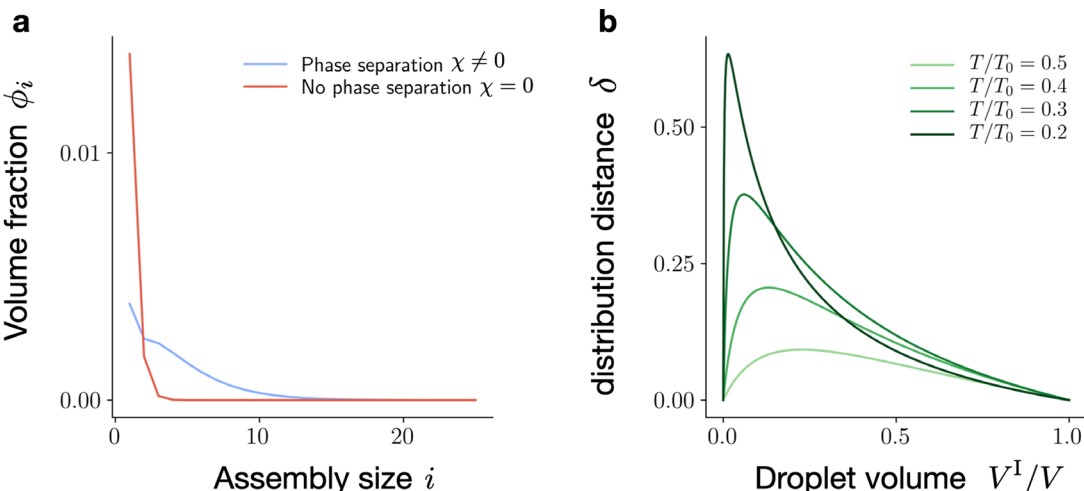

**Appendix 5—figure 2.** The influence of phase separation on assembly size. (**a**) Comparison between the size distribution in a homogeneous system and in the corresponding phase-separated system (averaged in both compartments). Here, we consider linear assemblies ($d = 1$), $M \to \infty$, $\phi_{tot} = 0.016$, and $T/T_0 = 0.2$. We note that the presence of compartments can favour assembly formation, even when the corresponding homogeneous mixture is populated mainly by monomers. The difference in distributions can be quantified utilizing the functional distance, defined in *Equation 35*. (**b**) The magnitude of this distance depends on the droplet size and the temperature chosen. The volume corresponding to the maximum distribution distance shifts towards lower values with decreasing temperature $T/T_0$. The distributions separated by the maximum distance, for $T/T_0 = 0.2$, are the ones displayed in **a**. $e_{int}/\chi = -0.5$, $s_{int}/k_B = -2$, $T/T_0 = 0.25$.

*Appendix 5—figure 1* clearly shows that the presence of assemblies affects the phase equilibrium of a mixture. We now prove that, in turn, the total number of assemblies can differ between phase-separating and homogeneous systems with the same total protein volume fraction. To show this, we fix the interaction propensity $\chi$, the temperature $T/T_0$, and the total macromolecule volume fraction $\phi_{tot}$ to values corresponding to two-phase coexistence at thermodynamic equilibrium. We then compare the assembly size distribution (after averaging over both phases) with the distribution in the corresponding homogeneous state, with the same values of $T$ and $\phi_{tot}$. Recalling that due to our choice of interaction propensity scaling in *Equation 9*, the size distribution in the homogeneous system, *Equation 4*, does not depend on $\chi$. For this reason, the homogeneous state can be thought of as an unstable state corresponding to the same $\chi$ as the phase separating one, which has not reached phase equilibrium yet, but also as the equilibrium state of a system with the same parameters as the phase separating one, but formed by assemblies that do not interact with the solvent ($\chi = 0$).

In **Appendix 5—figure 2a**, we display results for linear assemblies ($d = 1$) with $T/T_0 = 0.2$, and $\phi_{\text{tot}} = 0.016$. We compare the size distribution in the homogeneous system $\phi_i^{\text{h}}$, with the weighted average over compartments, defined as

$$\bar{\phi}_i = \frac{V^{\text{I}}}{V}\phi_i^{\text{I}} + \frac{V^{\text{II}}}{V}\phi_i^{\text{II}},\tag{34}$$

in the corresponding phase-separated system. Clearly, the two distributions differ, showing that the presence of compartments can lead to larger assemblies. The difference in size distributions can be quantified utilizing the so-called total variation distance, defined as

$$\delta(h, g) = \sup_i \left| \frac{h_i}{\sum_i h_i} - \frac{g_i}{\sum_i g_i} \right|.\tag{35}$$

This quantity characterizes the distance between two normalised functions as the largest possible distance among values that they assign to the same argument. The distance between the homogeneous size distribution and the distribution defined in **Equation 34** depends on the temperature $T$ and the total volume fraction $\phi_{\text{tot}}$, which in turn determines the droplet size. In **Appendix 5—figure 2b**, we display distribution distances $\delta(\phi^{\text{h}}, \bar{\phi})$ corresponding to different temperatures and droplet volumes. In the limits $V^{\text{I}}/V \to 0$ and $V^{\text{I}}/V \to 1$, the system becomes homogeneous. As a result, the distribution distance $\delta(\phi^{\text{h}}, \bar{\phi})$ vanishes. Note that the volume corresponding to the maximum distribution distance shifts towards lower values.

## Appendix 6

## A Assembly kinetics in homogeneous mixtures

In this section, we give the details on the kinetic theory for assembly in non-dilute homogeneous systems that can relax toward chemical equilibrium. Each component $i$ follows

$$\frac{d\phi_i}{dt} = r_i. \tag{36}$$

The assembly rates $r_i$ read

$$r_i = i\frac{1}{2}\sum_{j+k=i}\Delta r_{jk} - i\sum_j \Delta r_{ij}. \tag{37}$$

These rates conserve the total volume fraction $\phi_{\text{tot}}$, i.e., $\partial_t \phi_{\text{tot}} = \sum_i r_i = 0$. The assembly flux between two assemblies of size $i$ and $j$, and the combined $(i+j)$-mer reads

$$\Delta r_{ij} = k_{ij}\left[\exp\left(\frac{i\mu_i + j\mu_j}{k_{\text{B}}T}\right) - \exp\left(\frac{(i+j)\mu_{i+j}}{k_{\text{B}}T}\right)\right], \tag{38}$$

and is determined by differences in chemical potential per monomer. We now isolate the logarithmic part in the chemical potential and introduce chemical activities $\gamma_i$ via

$$\mu_i = i^{-1}\ln\left(\frac{\phi_i}{i}\gamma_i\right). \tag{39}$$

Choosing $k_{ij} = 2\tilde{k}\exp\left[-[i\ln\gamma_i + j\ln\gamma_j]\right]$, we can recast the assembly flux in *Equation 37* as

$$\Delta r_{ij} = 2\tilde{k}\left(\frac{\phi_i}{i}\frac{\phi_j}{j} - F_{ij}\frac{\phi_{i+j}}{i+j}\right), \tag{40}$$

which leads to a finite flux $\Delta r_{ij}$ in the limit $\phi_i \ll 1$. In the literature, $F_{ij}$ is known as a fragmentation kernel, and in our case, it reads $F_{ij} = \exp\left[(i+j)\ln\gamma_{i+j} - i\ln\gamma_i - j\ln\gamma_j\right]$. For linear assemblies $(d=1)$ belonging to Class 1, the fragmentation kernel is constant $F_{ij} = \tilde{\phi} = \exp\left(\frac{\Delta\omega}{k_{\text{B}}T} - 1\right)$ in agreement with standard polymerisation models (*Krapivsky et al., 2010*). For linear assemblies $(d=1)$ belonging to Class 2, $F_{ij} = F(\phi_{\text{tot}}) = \exp\left(\frac{\Delta\omega}{k_{\text{B}}T} - 1 - \frac{\chi-\chi'}{k_{\text{B}}T}(1-\phi_{\text{tot}})\right)$, i.e., the fragmentation kernel is still size independent but now depends on the total monomer volume fraction $\phi_{\text{tot}}$. Note that by making the kinetic rate coefficients corresponding to associations of small assemblies, i.e., $k_{ij}$ with $i,j \ll M$, explicitly dependent on the presence of large assemblies, our framework can be easily generalised to include primary and secondary nucleation (*Michaels et al., 2018*). Following again (*Krapivsky et al., 2010*), we can express the time evolution of $\phi_i(t)$ as follows:

$$\phi_i(t) = \phi_{\text{tot}}\left(1-a(t)\right)^2 a(t)^{i-1}, \tag{41}$$

$$a(t) = \frac{1 - \exp[-(\eta^{-1}-\eta)\phi_{\text{tot}}t]}{\eta^{-1} - \eta\exp[-(\eta^{-1}-\eta)\phi_{\text{tot}}t]}, \tag{42}$$

where $\eta$ is the following function of the fragmentation kernel $F(\phi_{\text{tot}})$:

$$\eta = \frac{1 + 2\frac{\phi_{\text{tot}}}{F} - \sqrt{1 + 4\frac{\phi_{\text{tot}}}{F}}}{2\frac{\phi_{\text{tot}}}{F}}. \tag{43}$$

# B Assembly kinetics in phase-separated systems

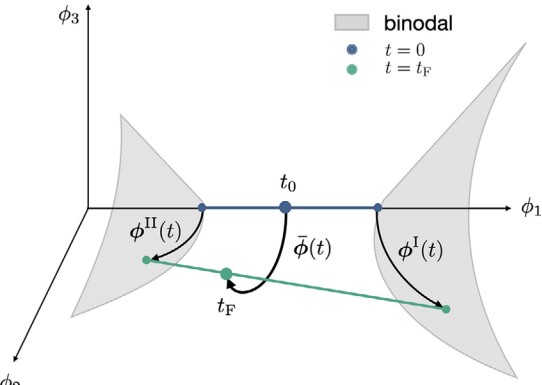

**Appendix 6—figure 1.** Kinetic trajectory in the multicomponent phase diagram. Illustration of the assembly kinetics at phase equilibrium for systems corresponding to $M = 3$ and initially composed of monomers only.

Here, we generalise the assembly kinetics described in the previous section to the case of phase coexistence. To this end, we focus on passive systems that can relax toward thermodynamic equilibrium. Moreover, we restrict ourselves to systems that are at phase equilibrium at any time during the relaxation kinetics toward thermodynamic equilibrium and following the theory originally developed in *Bauermann et al., 2022*. Chemical kinetics constrained to phase equilibrium is valid if the chemical reaction rates are small compared to diffusion rates. By choosing initial average volume fractions corresponding to two-phase coexistence, we can consider the system volume to be divided into two homogeneous compartments as a result of phase separation. We then study the time evolution of compartment sizes and volume fractions due to chemical reactions, enforcing instantaneous phase equilibrium at all times. To this aim, we start with the variation of particle numbers in compartments I and II:

$$\frac{dN_i^{\mathrm{I/II}}}{dt} = -J_i^{\mathrm{I/II}} + R_i^{\mathrm{I/II}}, \tag{44}$$

where $R_i^{\mathrm{I/II}}$ are the variations due to chemical reactions and $J_i^{\mathrm{I/II}}$ describes the exchange of assemblies between the two phases. Particle conservation during crossing implies $J_i^{\mathrm{I}} = -J_i^{\mathrm{II}}$. Due to volume conservation in the two-phase, we have

$$V^{\mathrm{I/II}} = \sum_{i=0}^{M} N_i^{\mathrm{I/II}} \nu_i. \tag{45}$$

Furthermore, $V = V^{\mathrm{I}} + V^{\mathrm{II}}$. We now introduce volume fractions $\phi_i^{\mathrm{I/II}} = N_i^{\mathrm{I/II}}/V^{\mathrm{I/II}}$ and the rescaled rates $j_i^{\mathrm{I/II}} = \nu_i J_i^{\mathrm{I/II}}/V^{\mathrm{I/II}}$ and $r_i^{\mathrm{I/II}} = \nu_i R_i^{\mathrm{I/II}}/V^{\mathrm{I/II}}$, leading to

$$\frac{\mathrm{d}\phi_i^{\mathrm{I/II}}}{\mathrm{d}t} = -j_i^{\mathrm{I/II}} + r_i^{\mathrm{I/II}} - \phi_i^{\mathrm{I/II}} \frac{\mathrm{d}\ln V^{\mathrm{I/II}}}{\mathrm{d}t}, \tag{46}$$

which correspond to *Equation 36* generalised to two-phase coexistence. The rates in both phases $r_i^{\mathrm{I/II}}$ are given in *Equation 18; Equation 44; Equation 45* can be combined to get $(d/dt)V^{\mathrm{I/II}} = V^{\mathrm{I/II}} \sum_{i=0}^{M}(r_i^{\mathrm{I/II}} - j_i^{\mathrm{I/II}})$. Using the volume-conserving properties of the rates, $\sum_{i=1}^{M} r_i^{\mathrm{I/II}} = 0$, we finally obtain

$$\frac{\mathrm{d}\ln V^{\mathrm{I/II}}}{\mathrm{d}t} = -\sum_{i=0}^{M} j_i^{\mathrm{I/II}}. \tag{47}$$

Assembly mass conservation at the interface implies

$$j_i^{\mathrm{I}} = -j_i^{\mathrm{II}} \frac{V^{\mathrm{II}}}{V^{\mathrm{I}}}, \tag{48}$$

with the volume dynamics obeying $(d/dt)(V^{\mathrm{I}} + V^{\mathrm{II}}) = 0$.

The currents $j_i^{\mathrm{I/II}}$ enforce that phase equilibrium is satisfied at all times, which can be expressed by taking a time derivative of *Equation 7a*, *Equation 7b*:

$$\sum_{j=1}^{M} \frac{\partial \mu_i^{\mathrm{I}}}{\partial \phi_j^{\mathrm{I}}} \frac{d\phi_j^{\mathrm{I}}}{dt} = \sum_{j=1}^{M} \frac{\partial \mu_i^{\mathrm{II}}}{\partial \phi_j^{\mathrm{II}}} \frac{d\phi_j^{\mathrm{II}}}{dt}, \tag{49a}$$

$$\sum_{j=1}^{M} \frac{\partial \Pi_i^{\mathrm{I}}}{\partial \phi_j^{\mathrm{I}}} \frac{d\phi_j^{\mathrm{I}}}{dt} = \sum_{j=1}^{M} \frac{\partial \Pi_i^{\mathrm{II}}}{\partial \phi_j^{\mathrm{II}}} \frac{d\phi_j^{\mathrm{II}}}{dt}, \tag{49b}$$

provided that the initial phase volume and volume fractions $V^{\mathrm{I}}(t=0)$, and $\phi_i^{\mathrm{I/II}}(t=0)$ are a solution of *Equation 7a*, *Equation 7b*. Once an expression for $\partial \mu_i/\partial \phi_j$ and $\partial \Pi/\partial \phi_j$ is calculated, we can derive a set of $(M+1)$ equations for $j_i^{\mathrm{I}}$ inserting *Equations 46–48* in *Equation 49a*. These equations are linear and enable us to find an expression for $j_i^{\mathrm{I/II}}$ as a function of $\phi_i^{\mathrm{I/II}}$ and $V^{\mathrm{I}}/V$. We have finally all the ingredients to characterize the dynamics of the phase volume and volume fractions $\phi_i^{\mathrm{I/II}}(t)$ and $V^{\mathrm{I}}(t)$, integrating *Equation 46; Equation 47* and provided we can solve the initial phase equilibrium problem to find $V^{\mathrm{I}}(t=0)/V$, and $\phi_i^{\mathrm{I/II}}(t=0)$. This scheme can be used to study the kinetics of a system initially composed of two phases filled by monomers only that relax to its thermodynamic equilibrium. An illustration of such relaxation kinetics is depicted in *Appendix 6—figure 1*. Note that the currents $j_i^{\mathrm{I/II}}$ restrict the trajectories to lie in the binodal manifold at all times.

