## [Editor Report · eLife Assessment]

The authors present an **important** theoretical framework that describes the interplay between liquid-liquid phase separation and protein aggregation within a mean-field model. This work will be of high interest to the biophysics and molecular biology communities, as it will help understand and analyse assembly within biomolecular condensates in cells or in-vitro. Major strengths of this **convincing** work are the consideration of aggregates with various dimensionality and the possibility for protein gelation.

---

## [Referee Report · Reviewer #3 (Public review)]

Summary:

The authors combine classical theories of phase separation and self-assembly to establish a framework for explaining the coupling between the two phenomena in the context of protein assemblies and condensates. By starting from a mean-field free energy for monomers and assemblies immersed in solvent and imposing conditions of equilibrium, the authors derive phase diagrams indicating how assemblies partition into different condensed phases as temperature and the total volume fraction of proteins are varied. They find that phase separation can promote assembly within the protein-rich phase, providing a potential mechanism for spatial control of assembly. They extend their theory to account for the possibility of gelation. They also create a theory for the kinetics of self-assembly within phase separated systems, predicting how assembly size distributions change with time within the different phases as well as how the volumes of the different phases change with time.

Review For Revision:

The revised manuscript provides better motivation and physical explanations for the equations, and the authors have addressed references, typos, and other minor technical issues identified in the review. These changes have significantly improved the manuscript.

---

## [Author Response]

The following is the authors’ response to the original reviews.

**Public Reviews:**

**Reviewer 1 (Public Review):**
Summary:The authors present a mean-field model that describes the interplay between (protein) aggregation and phase separation. Different classes of interaction complexity and aggregate dimensionality are considered, both in calculations concerning (equilibrium) phase behavior and kinetics of assembly formation.Strengths:The present work is, although purely theoretical, of high interest to understanding biological processes that occur as a result of a coupling between protein aggregation and phase separation. Of course, such processes are abundant, in the living cell as well as in in-vitro experiments. I appreciate the consideration of aggregates with various dimensionality, as well as the categorization into different ”interaction classes”, together with the mentioning of experimental observations from biology. The model is convincing and underlines the complexity associated with the distribution of proteins across phases and aggregates in the living cell.Weaknesses:There are a few minor weaknesses.
**Reviewer 2 (Public Review):**
This work deals with a very difficult physical problem: relating the assembly of building blocks on a molecular scale to the appearance of large, macroscopic assemblies. This problem is particularly difficult to treat, because of the large number of units involved, and of the complex way in which these units-monomers-interact with each other and with the solvent. In order to make the problem treatable, the authors recur to a number of approximations: Among these, there is the assumption that the system is spatially homogeneous, i.e., its features are the same in all regions of space. In particular, the homogeneity assumption may not hold in biologically relevant systems such as cells, where the behavior close to the cell membrane may strongly differ from the one in the bulk. As a result, this hypothesis calls for a cautious consideration and interpretation of the results of this work. Another notable simplification introduced by the authors is the assumption that the system can only follow two possible behaviors: In the first, each monomer interacts equally with the solvent; no matter the size of the cluster of which it is part. In the second case, monomers in the bulk of a cluster and monomers at the assembly boundary interact with the solvent in a different way. These two cases are considered not only because they simplify the problem, but also because they are inspired by biologically relevant proteins.With these simplifications, the authors trace the phase diagram of the system, characterizing its phases for different fractions of the volume occupied by the monomers and solvent, and for different values of the temperature. The results qualitatively reproduce some features observed in recent experiments, such as an anomalous distribution of cluster sizes below the system saturation threshold, and the gelation of condensed phases above such threshold.
**Reviewer 3 (Public Review):**
Summary:The authors combine classical theories of phase separation and self-assembly to establish a framework for explaining the coupling between the two phenomena in the context of protein assemblies and condensates. By starting from a mean-field free energy for monomers and assemblies immersed in solvent and imposing conditions of equilibrium, the authors derive phase diagrams indicating how assemblies partition into different condensed phases as temperature and the total volume fraction of proteins are varied. They find that phase separation can promote assembly within the protein-rich phase, providing a potential mechanism for spatial control of assembly. They extend their theory to account for the possibility of gelation. They also create a theory for the kinetics of self-assembly within phase separated systems, predicting how assembly size distributions change with time within the different phases as well as how the volumes of the different phases change with time.Strengths:The theoretical framework that the authors present is an interesting marriage of classic theories of phase separation and self-assembly. Its simplicity should make it a powerful general tool for understanding the thermodynamics of assembly coupled to phase separation, and it should provide a useful framework for analyzing experiments on assembly within biomolecular condensates.The key advance over previous work is that the authors now account for how self-assembly can change the boundaries of the phase diagram.A second interesting point is the explicit theoretical consideration for the possibility that gelation (i.e. self-assembly into a macroscopic aggregate) could account for widely observed solidification of condensates. While this concept has been broadly discussed, to date I have yet to see a rigorous theoretical analysis of the possibility.The kinetic theory in sections 5 and 6 is also interesting as it extends on previous work by considering the kinetics of phase separation as well as those of self-assembly.Weaknesses:A key point the authors make about their theory is that it allows, as opposed to previous research, to study non-dilute limits. It is true that they consider gelation when the 3D assemblies become macroscopic. However, dilute solution theory assumptions seem to be embedded in many aspects of their theory, and it is not always clear where else the non-dilute limits are considered. Is it in the inter-species interaction *χij*? Why then do they never explore cases for which *χij* is nonzero in their analysis?

We explicitly consider that monomers and aggregates are non-dilute with respect to solvent. This is evident in accounting for the mixing entropy of all components, including the solvent. Moreover, we account for interactions among the monomers and the different aggregates with the solvent. We consider the case where each monomeric unit, independent in aggregate it is part of, interacts the same way with the solvent. Please note that this case corresponds to a non-dilute scenario where interactions indeed drive phase separation.

The connection between this theory and biological systems is described in the introduction but lost along the main text. It would be very helpful to point out, for instance, that the presence of phase separation might induce aggregation of proteins. This point is described formally at the end of Section 3, but a more qualitative connection to biological systems would be very useful here.

We thank the referee for the useful comment, we now mention this in the introduction (line 80) and point out the biological relevance of assembly formation and localization via the presence of phase separation (lines 268 and 283).

Building on the previous point, it would be helpful to give an intuitive sense of where the equations derived in the Appendices and presented in the main text come from and to spell out clear physical interpretations of the results. For example, it would be helpful to point out that Eq. 4 is a form of the law of mass action, familiar from introductory chemistry. It would be useful to better explain how the current work extends on existing previous work from these authors as well as others. Along these lines, closely related work by W. Jacobs and B. Rogers [O. Hedge et al. 2023, https://arxiv.org/abs/2301.06134; T. Li et al. 2023, https://arxiv.org/abs/2306.13198] should be cited in the introduction. The results discussed in the first paragraph of Section 3 on assembly size distributions in a homogeneous system are well-known from classic theories of self-assembly. This should be acknowledged and appropriate references should be added; see for instance, Rev. Mod. Phys. 93, 025008 and Statistical Thermodynamics Of Surfaces, Interfaces, And Membranes by Sam Safran. Equation 14 for the kinetic of volume fractions is given with reference to Bauermann et al. 2022, but it should be accompanied by a better intuitive interpretation of its terms in the main text. In particular, how should one understand the third term in this equation? Why does the change in volume impact the change of volume fraction in this way?

We thank the referee for the suggestions. We have included the missing references, with a particular emphasis on DNA nanostars that inhibit phase separation in DNA liquids in the definition of class II. We added intuitive explanations of the main equations, such as Eqs. (4),(8),(14), (17), and (18). Notice that, according to Mysels, Karol J., J. Chem. Educ., 33, 178 (1956) (https://pubs-acs-org.sire.ub.edu/doi/epdf/10.1021/ed033p178) we refer to (18) as the law of mass action.

The discussion in the last paragraph of Section 6 should be clarified. How can the total amount of protein in both phases decrease? This would necessarily violate either mass or volume conservation. Also, the discussion of why the volume is non-monotonic in time is not clear.

A decrease in the total amount of protein in both phases does not violate mass conservation, if the volume of the phases varies accordingly. In particular, the volume of the denser phase should grow. This given, in the case presented the total protein amount in the dense phase decreases, while in the dilute phase increases. For this reason, we revised the paragraph and now explain the results in more detail (see lines starting from 407). The nonmonotonic volume change is indeed a puzzling finding that, as we now state in the manuscript, requires further investigation. Given the lack of analytical approaches available to tackle the complex kinetics in the presence of coexisting phases, we believe that this analysis goes beyond the scope of the present paper.

**Recommendations for the authors**

**Reviewer 1 (Recommendations For The Authors):**
Line 96: I feel a mentioning/definition/explanation and perhaps some discussion on the parameter M (limiting aggregate size) would have been in place in the introduction of Equation (1). Furthermore, in the usual interpretation, Flory interaction parameters (symbolized *χ*) are dimensionless, as, classically, they represent an exchange energy (normalized by kT), defined on a monomeric basis. Here they seem to carry the dimension of energy.

We thank the reviewer for the observation. We have included a brief comment on M and mentioned that we use *χ* parameters that carry the dimension of energy such that, varying *kBT*, we scale at the same time the term containing interaction propensities (*χ*) and the one containing internal energies (_e_int). See the comment on line 127

Line 150: The choice of *ρi* = *i* physically implies that a single protein is assumed to have the same as a solvent molecule. This may be a bit of a stretch. This assumption leads to an overestimation of the translational entropy of the aggregates (first term in Equation (1)). Acknowledging that _ρ_1 *>> ρs* would give a pronounced desymmetrization of the phase diagram (I suspect).

Indeed, in the case of monomers only, the assumption leads to a symmetric phase diagram which may be unrealistic. Once assemblies form, however, the phase diagram becomes asymmetric and for this reason we decided to assume *ρi* = *i*, simplifying the theoretical analysis. We have added a clarifying sentence in the manuscript, see line 163

Furthermore, the pictures in Figure 1a-c suggest the presence of a disordered residue, the degree of swelling of which might affect binding strength (see for instance: https://doi.org/10.3389/fnmol.2022.962526).

We added a comment on the possible coupling between internal free energies and interaction propensities, such as the swelling mechanism that affects binding sites, and included the reference above (line 215).

Line 154-156: It’s unclear what is meant with ”an internal bond that keeps each assembly together”. How should this be interpreted on an intuitive physical level?

We apologise for being unclear. We meant the internal bonds that lead to the formation of assemblies. We have now rephrased this sentence in the main text (lines starting from 169).

Line 254: The fact that *ϕsg* is defined below does not mean it does not fall out of the air here. The same holds for the consideration of the limit *M* →∞. Ideally, the main text should stand on its own, in particular with respect to physical intuitiveness, as well as the necessity and interest of discussion topics. Technical details, derivations and additional information can be in an appendix.

We agree with the referee and added some physical insights about the limit. We now also state clearly in the main text (line 298) that _ϕ_sg is affected by temperature and the free energy of internal bonds.

Line 257: ”Since we do not explicitly include the solvent in assembly formation we will consider the gel as a phase without solvent and thus *ϕtot* = 1”. I’m not sure if I can agree with this. I would say, a gel, certainly in biological context, almost per definition contains a large fraction of solvent, i.e. here water. The situation ”*ϕtot* = 1” would rather be a solid precipitate. Is gelation properly captured by this model?

We thank the referee for this very relevant observation. We now state in the main text that the model predicts a macroscopic assembly which we call ’the gel phase’, in agreement with previous literature. Then, to clarify, we added the sentence ”Please note that, since we do not explicitly include the solvent in assembly formation (see reaction scheme in Fig.1a), in our model the gel corresponds to a phase without solvent, _ϕ_tot = 1. To account for biological gels that can be rich in water, our theory can be straightforwardly extended by incorporating the solvent into the reaction scheme.”, see main text line 300.

Line 268: Shouldn’t ”solvent” be ”solution”? If *fsol* is given by Equation (1), surely not only the solvent is considered.

Indeed, this is a typo, and we now use the term ’solution’ instead of ’solvent.’

Line 273: At this stage, the only information provided in the main text is that *ω*∞ is ”a constant that does not affect chemical nor phase equilibrium, except in the limit *M* →∞” (see lines 153-154). This is a little bit too abstract for me. Again, the main text should stand on its own, meaning the reader should not have to rely on an appendix to at least have an intuitive physical understanding of any modeling or input parameter discussed in the main text.

We thank the reviewer for pointing this out. We now comment on the physical interpretation of *ω*∞ in the main text, see lines from 320 on.

Figure 4. \begin{document}$\tilde{k}$\end{document} appears in Equation (39) but it is not defined.

We thank the reviewer for pointing this out. We have reshaped appendix 6A, making use of chemical activities and clarified the origin of the rate \begin{document}$\tilde{k}$\end{document}.

Line 317. I don’t fully understand the intention of the remark on the model being adaptable for ”primary and secondary nucleation”. How/in what way is this different from association and dissociation? For instance, classical nucleation theory is based on association and dissociation of monomeric units to and from clusters.

We agree that the kinetic rate coefficients *kij* (appearing in the association and dissociation rates ∆*rij*, Eq. 17) in our manuscript already depend on assembly length, see Appendix 6 B, where we now clarified their definition. Please note that, however, that secondary nucleation is a special kind of association, for which the kinetic rate coefficients corresponding to associations of small assemblies, i.e. *kij* with_i,j_ ≪ *M*, explicitly depend on the presence of large assemblies with sizes *l* ≫ 1. In our manuscript, we have not accounted for such a dependence. We now make this aspect clear in the manuscript, see Appendix 6 B.

Line 321. Why is ∆*rij* called the ”monomer exchange rate”? In line 318 the same parameter is defined as the ”reaction rate for the formation of a (i+j)-mer”. Why should these be the same?

We thank the reviewer for spotting this typo.

Line 323. Why do these calculations use M = 15?

The exploration of a 15-dimensional phase space is already numerically challenging. We are currently working on a generalization of the numerical scheme to work with larger values of M but, to discuss the fundamental physical principles, we kept *M* = 15.

**Reviewer 2 (Recommendations For The Authors):**
The manuscript presents several issues, on both the scientific and presentational level, which need to be carefully addressed. Please find below a list of the points that need to be addressed by the authors, divided into major and minor points. Major issues:• A general, major concern about the results in the paper is the homogeneity assumption. I do understand that repeating the whole analysis presented in the manuscript by allowing for spatial inhomogeneities partially goes beyond the scope of this paper. However, the authors should at least discuss how such inhomogeneities may alter the results in a qualitative way, and treat explicitly the presence of inhomogeneity in one prototypical case treated in the manuscript. Namely, what happens if the volume fractions and relative molecular volumes in the free energy (1) depend on space, e.g., *ϕi* → *ϕi*(*x*)?

We would like to stress that, in the present paper, we do account for spatial inhomogeneities. Indeed, in the case of phase separation, we consider systems which are divided into two phases, characterized by different values of the assemblies’ volume fractions *ϕi*. We do, however, consider the system to be homogeneous inside the phases, implying a jump in the value of the volume fraction at the interface between the two phases. In this sense, the analysis we carry out is valid in the thermodynamic limit, where gradients of the volume fractions *ϕi*(*x*) within the phases, can be neglected. On the other hand, considering the full spatial problem, i.e. solving the equations for *M* = 15 spatially varying fields, would be numerically extremely challenging.

• The authors’ results relate molecular assembly- a phenomenon at the molecular scale-to phase separation-a mesoscopic or macroscopic phenomenon. The authors should stress the conceptual importance of this connection between scales, and present their results from the perspective of a multi-scale model.

We thank the reviewer for pointing this out. We now emphasize the multi-scale feature of our model in the introduction (line 80).

• Starting from Section 1, the reader is not well guided through the sections that follow. The authors should provide an outline of the line of though that they are going to follow in the following sections, and logically connect each section to the next one with a short paragraph at the end of each section. This paragraph should resume what has been addressed in the current section, and the connection with the topic that will be addressed in the next one.

We agree with the reviewer and have added a transitioning sentence at the end of each paragraph.

• ’We focus on linear assemblies (d = 1)’: Given the striking differences of the results between d = 1 and *d >* 1 shown above, the authors should discuss what happens for *d >* 1 as well.• ’In figure Fig. 5a, we show the initial and final equilibrium binodals (black and coloured curve, respectively), for the case of linear assemblies (d = 1) belonging to class 1’: Again, show what happens for *d >* 1.

We agree with the reviewer, the kinetics in *d >* 1 would be definitely interesting. However, in this case, one assembly can become macroscopic (i.e. *M* must be set to ∞). This requires some substantial modification in the kinetic scheme, like introducing an absorbing boundary condition for monomers ’sucked in’ the gel. We prefer to leave this for future work, and now state it explicitly in the manuscript (line 383).

• ’This difference arises because, within class 2, monomers in the bulk of an assembly have reduced interaction propensity with respect to the boundary ones. As a consequence, the formation of large clusters shifts the onset of phase separation to higher *ϕtot* values.’: To prove this argument, the authors should show Fig. 2g and h for *d >* 1. In fact, by varying d, the effect of the boundary vs. bulk also varies.

We prefer to discuss the thermodynamics of *d >* 1 in section 4 on gelation. There we present only a single phase diagram so as not to blow up the discussion on equilibrium too much.

• ’referring for simplicity to systems belonging to Class 1’: The authors should do the same analysis for Class 2.

We agree with the reviewer. However, again not to blow up the discussion on equilibrium, we leave it for future work.

• ’other, implying that the corresponding Flory-Huggins parameter *χij* vanishes’: Why?

The explanation based on a lattice model is reported in Appendix 2, and is now more clearly referenced (line 185).

Minor issues:• Eq. (10): Here the authors should explain in the main text, possibly in a simple and intuitive way, why the number of monomers i and the space dimension d enter the righthand side of this equation in this particular way.

We thank the reviewer for pointing this out. We added the physical origin of the scaling with dimension in Eq. (10) and in Eq. (8), as pointed out by reviewer 3.

• ’The second and fifth terms of *fsol* characterize the internal free energies’: What do you mean by ’characterize the internal free energies’? Please clarify.

As we now state more clearly (lines 114-120), these two contributions include the internal free energies _ω_s and *ωi*, stemming from the free energy of internal bonds that lead to assembly formation.

• ’depend on the scaling form of the’: Scaling with respect to what ? Please clarify.

We have now clarified that the scaling is with respect to the assembly size *i*.

• Figure 2 is way too dense: it should be split into two figures, and the legend of each of the two figures should be expanded to properly guide the reader to understand the figures.

We understand the reviewer’s point of view. To avoid altering the present flow, we decided not to split the figure, but we have included shaded boxes to better guide the reader.

• ’this is a consequence of the gelation transition’: Please clarify• ’and this limitation can be dealt with by introducing explicitly the infinite-sized gel in the free energy’: Why? Please clarify.

We have now rephrased these sentences, hopefully in a clearer way. We now state: ’We know that this divergence is physical, and is caused by the gelation transition. This limitation can be dealt with by introducing explicitly a term in the free energy that accounts for an infinite-sized assembly (the gel)’, see lines 320-322.

• Figure 4: Add plots of panels d, e, h and i with log scale on the y axis to make explicit an eventual exponential behavior, and revise the text accordingly

Not to further complicate Figure 4, we preferred to display the logarithmic plots of the equilibrium distribution in the appendix, see Figure A3-1.

• ’... an equilibrium distribution which monotonously decreases with assembly size’: It is not the distributions that decreases but the cluster volume fraction, please rephrase.

We thank the reviewer for pointing this out and have now rephrased this sentence (line 394).

**Reviewer 3 (Recommendations For The Authors):**
I could not obtain the exact form of Eq 29 in App 3, can the authors elaborate on this calculation. App 3: What does it mean binodal agrees well with *ϕsg*? And doesn’t *ϕsg* depend on temperature through phi tilde? What temperature is this result for?

We apologise for the unclear explanation. We now state in detail that Eq. (29) is obtained by plugging the expression of *ϕi* given in Eq. (24) into Eq. (1), in the main text. The dependence of *ϕ*_1_ on *ϕ*_tot_ is expressed in Eq. (26), and we have omitted linear terms in *ϕ*_tot_, since they do not affect phase equilibrium (see lines 802-809). Moreover, *ϕ*sg depends indeed on *kBT*. We refer to the comparison between the full curve *ϕ*sg in the *kBT−ϕ*_tot_ plane, and the branch of the binodal between the triple point (indicated now with a cross) and *ϕ*_tot_ = 1. The two curves are close, as expected since both correspond to the boundary between homogeneous mixtures and the gel state, obtained with different methods.

The references to Figures in the appendices are confusing. Please make it clear whether Figures in the main text or the appendices are being referenced. On a related note, the Appendix figures seem to be placed in appendices whose text describes something else - Appendix 2, Figure 1 should be moved to Appendix 3; Appendix 3, Figure 1 should be moved to Appendix 4; etc.

We revised the appendix, corrected the figure positions and clarified their references.